# Imitation-Projected Programmatic Reinforcement Learning

**Abhinav Verma**[*]
Rice University
averma@rice.edu

**Hoang M. Le**[*]
Caltech
hmle@caltech.edu

**Yisong Yue**
Caltech
yyue@caltech.edu

**Swarat Chaudhuri**
Rice University
swarat@rice.edu

## Abstract

We study the problem of programmatic reinforcement learning, in which policies are represented as short programs in a symbolic language. Programmatic policies can be more interpretable, generalizable, and amenable to formal verification than neural policies; however, designing rigorous learning approaches for such policies remains a challenge. Our approach to this challenge — a meta-algorithm called PROPEL— is based on three insights. First, we view our learning task as optimization in policy space, modulo the constraint that the desired policy has a programmatic representation, and solve this optimization problem using a form of mirror descent that takes a gradient step into the unconstrained policy space and then projects back onto the constrained space. Second, we view the unconstrained policy space as mixing neural and programmatic representations, which enables employing state-of-the-art deep policy gradient approaches. Third, we cast the projection step as program synthesis via imitation learning, and exploit contemporary combinatorial methods for this task. We present theoretical convergence results for PROPEL and empirically evaluate the approach in three continuous control domains. The experiments show that PROPEL can significantly outperform state-of-the-art approaches for learning programmatic policies.

## 1 Introduction

A growing body of work [58, 8, 60] investigates reinforcement learning (RL) approaches that represent policies as programs in a symbolic language, e.g., a domain-specific language for composing control modules such as PID controllers [5]. Short programmatic policies offer many advantages over neural policies discovered through deep RL, including greater interpretability, better generalization to unseen environments, and greater amenability to formal verification. These benefits motivate developing effective approaches for learning such programmatic policies.

However, programmatic reinforcement learning (PRL) remains a challenging problem, owing to the highly structured nature of the policy space. Recent state-of-the-art approaches employ program synthesis methods to imitate or distill a pre-trained neural policy into short programs [58, 8]. However, such a distillation process can yield a highly suboptimal programmatic policy — i.e., a large distillation gap — and the issue of direct policy search for programmatic policies also remains open.

In this paper, we develop PROPEL (Imitation-**Pro**jected **P**rogrammatic R**e**inforcement **L**earning), a new learning meta-algorithm for PRL, as a response to this challenge. The design of PROPEL is based on three insights that enables integrating and building upon state-of-the-art approaches for policy gradients and program synthesis. First, we view programmatic policy learning as a constrained policy optimization problem, in which the desired policies are constrained to be those that have a programmatic representation. This insight motivates utilizing constrained mirror descent approaches, which take a gradient step into the unconstrained policy space and then project back onto the constrained space. Second, by allowing the unconstrained policy space to have a mix of neural

---

[*]Equal contribution

$$\pi(s) \quad ::= \quad a \mid Op(\pi_1(s), \ldots, \pi_k(s)) \mid \textbf{if } b \textbf{ then } \pi_1(s) \textbf{ else } \pi_2(s) \mid \oplus_\theta(\pi_1(s), \ldots, \pi_k(s))$$
$$b \quad ::= \quad \phi(s) \mid BOp(b_1, \ldots, b_k)$$

Figure 1: *A high-level syntax for programmatic policies, inspired by [58]. A policy $\pi(s)$ takes a state $s$ as input and produces an action $a$ as output. $b$ represents boolean expressions; $\phi$ is a boolean-valued operator on states; $Op$ is an operator that combines multiple policies into one policy; $BOp$ is a standard boolean operator; and $\oplus_\theta$ is a "library function" parameterized by $\theta$.*

$$\textbf{if } (s[\texttt{TrackPos}] < 0.011 \textbf{ and } s[\texttt{TrackPos}] > -0.011)$$
$$\textbf{then } \texttt{PID}_{\langle \text{RPM}, 0.45, 3.54, 0.03, 53.39 \rangle}(s) \textbf{ else } \texttt{PID}_{\langle \text{RPM}, 0.39, 3.54, 0.03, 53.39 \rangle}(s)$$

Figure 2: *A programmatic policy for acceleration in* TORCS *[59], automatically discovered by* PROPEL. $s[\texttt{TrackPos}]$ *represents the most recent reading from sensor* TrackPos.

and programmatic representations, we can employ well-developed deep policy gradient approaches [55, 36, 47, 48, 19] to compute the unconstrained gradient step. Third, we define the projection operator using program synthesis via imitation learning [58, 8], in order to recover a programmatic policy from the unconstrained policy space. Our contributions can be summarized as:

- We present PROPEL, a novel meta-algorithm that is based on mirror descent, program synthesis, and imitation learning, for PRL.

- On the theoretical side, we show how to cast PROPEL as a form of constrained mirror descent. We provide a thorough theoretical analysis characterizing the impact of approximate gradients and projections. Further, we prove results that provide expected regret bounds and finite-sample guarantees under reasonable assumptions.

- On the practical side, we provide a concrete instantiation of PROPEL and evaluate it in three continuous control domains, including the challenging car-racing domain TORCS [59]. The experiments show significant improvements over state-of-the-art approaches for learning programmatic policies.

## 2  Problem Statement

The problem of programmatic reinforcement learning (PRL) consists of a Markov Decision Process (MDP) $\mathcal{M}$ and a programmatic policy class $\Pi$. The definition of $\mathcal{M} = (\mathcal{S}, \mathcal{A}, P, c, p_0, \gamma)$ is standard [54], with $\mathcal{S}$ being the state space, $\mathcal{A}$ the action space, $P(s'|s, a)$ the probability density function of transitioning from a state-action pair to a new state, $c(s, a)$ the state-action cost function, $p_0(s)$ a distribution over starting states, and $\gamma \in (0, 1)$ the discount factor. A policy $\pi : \mathcal{S} \to \mathcal{A}$ (stochastically) maps states to actions. We focus on continuous control problems, so $\mathcal{S}$ and $\mathcal{A}$ are assumed to be continuous spaces. The goal is to find a programmatic policy $\pi^* \in \Pi$ such that:

$$\pi^* = \underset{\pi \in \Pi}{\text{argmin}}\, J(\pi), \qquad \text{where: } J(\pi) = \mathbf{E}\left[\sum_{i=0}^{\infty} \gamma^i c(s_i, a_i \equiv \pi(s_i))\right], \qquad (1)$$

with the expectation taken over the initial state distribution $s_0 \sim p_0$, the policy decisions, and the transition dynamics $P$. One can also use rewards, in which case (1) becomes a maximization problem.

**Programmatic Policy Class.** A programmatic policy class $\Pi$ consists of policies that can be represented parsimoniously by a (domain-specific) programming language. Recent work [58, 8, 60] indicates that such policies can be easier to interpret and formally verify than neural policies, and can also be more robust to changes in the environment.

In this paper, we consider two concrete classes of programmatic policies. The first, a simplification of the class considered in Verma et al. [58], is defined by the modular, high-level language in Figure 1. This language assumes a library of parameterized functions $\oplus_\theta$ representing standard controllers, for instance Proportional-Integral-Derivative (PID) [6] or bang-bang controllers [11]. Programs in the language take states $s$ as inputs and produce actions $a$ as output, and can invoke fully instantiated library controllers along with predefined arithmetic, boolean and relational operators. The second, "lower-level" class, from Bastani et al. [8], consists of decision trees that map states to actions.

**Example.** Consider the problem of learning a programmatic policy, in the language of Figure 1, that controls a car's accelerator in the TORCS car-racing environment [59]. Figure 2 shows a program in our language for this task. The program invokes PID controllers $\texttt{PID}_{\langle j, \theta_P, \theta_I, \theta_D \rangle}$, where $j$ identifies

---

**Algorithm 1** Imitation-Projected Programmatic Reinforcement Learning (PROPEL)

---
1: **Input:** Programmatic & Neural Policy Classes: $\Pi$ & $\mathcal{F}$.
2: **Input:** Either initial $\pi_0$ or initial $f_0$
3: Define joint policy class: $\mathcal{H} \equiv \Pi \oplus \mathcal{F}$       *//h $\equiv \pi + f$ defined as $h(s) = \pi(s) + f(s)$*
4: **if** given initial $f_0$ **then**
5:     $\pi_0 \leftarrow$ PROJECT$(f_0)$       *//program synthesis via imitation learning*
6: **end if**
7: **for** $t = 1, \ldots, T$ **do**
8:     $h_t \leftarrow$ UPDATE$_\mathcal{F}(\pi_{t-1}, \eta)$       *//policy gradient in neural policy space with learning rate $\eta$*
9:     $\pi_t \leftarrow$ PROJECT$_\Pi(h_t)$       *//program synthesis via imitation learning*
10: **end for**
11: **Return:** Policy $\pi_T$

---

the sensor (out of 29, in our experiments) that provides inputs to the controller, and $\theta_P$, $\theta_I$, and $\theta_D$ are respectively the real-valued coefficients of the proportional, integral, and derivative terms in the controller. We note that the program only uses the sensors `TrackPos` and `RPM`. While `TrackPos` (for the position of the car relative to the track axis) is used to decide which controller to use, only the `RPM` sensor is needed to calculate the acceleration.

**Learning Challenges.** Learning programmatic policies in the continuous RL setting is challenging, as the best performing methods utilize policy gradient approaches [55, 36, 47, 48, 19], but policy gradients are hard to compute in programmatic representations. In many cases, $\Pi$ may not even be differentiable. For our approach, we only assume access to program synthesis methods that can select a programmatic policy $\pi \in \Pi$ that minimizes imitation disagreement with demonstrations provided by a teaching oracle. Because imitation learning tends to be easier than general RL in long-horizon tasks [53], the task of imitating a neural policy with a program is, intuitively, significantly simpler than the full programmatic RL problem. This intuition is corroborated by past work on programmatic RL [58], which shows that direct search over programs often fails to meet basic performance objectives.

## 3   Learning Algorithm

To develop our approach, we take the viewpoint of (1) being a constrained optimization problem, where $\Pi \subset \mathcal{H}$ resides within a larger space of policies $\mathcal{H}$. In particular, we will represent $\mathcal{H} \equiv \Pi \oplus \mathcal{F}$ using a mixing of programmatic policies $\Pi$ and neural polices $\mathcal{F}$. Any mixed policy $h \equiv \pi + f$ can be invoked as $h(s) = \pi(s) + f(s)$. In general, we assume that $\mathcal{F}$ is a good approximation of $\Pi$ (i.e., for each $\pi \in \Pi$ there is some $f \in \mathcal{F}$ that approximates it well), which we formalize in Section 4.

We can now frame our constrained learning problem as minimizing (1) over $\Pi \subset \mathcal{H}$, that alternate between taking a gradient step in the general space $\mathcal{H}$ and projecting back down onto $\Pi$. This "lift-and-project" perspective motivates viewing our problem via the lens of mirror descent [40]. In standard mirror descent, the unconstrained gradient step can be written as $h \leftarrow h_{prev} - \eta \nabla_\mathcal{H} J(h_{prev})$ for step size $\eta$, and the projection can be written as $\pi \leftarrow \operatorname{argmin}_{\pi' \in \Pi} D(\pi', h)$ for divergence measure $D$.

Our approach, *Imitation-Projected Programmatic Reinforcement Learning* (PROPEL), is outlined in Algorithm 1 (also see Figure 3). PROPEL is a meta-algorithm that requires instantiating two subroutines, UPDATE and PROJECT, which correspond to the standard update and projection steps, respectively. PROPEL can be viewed as a form of functional mirror descent with some notable deviations from vanilla mirror descent.

**UPDATE$_\mathcal{F}$.**   Since policy gradient methods are well-developed for neural policy classes $\mathcal{F}$ (e.g., [36, 47, 48, 30, 24, 19]) and non-existent for programmatic policy classes $\Pi$, PROPEL is designed to leverage policy gradients in $\mathcal{F}$ and avoid policy gradients in $\Pi$. Algorithm 2 shows one instantiation of UPDATE$_\mathcal{F}$. Note that standard mirror descent takes unconstrained gradient steps in $\mathcal{H}$ rather than $\mathcal{F}$, and we discuss this discrepancy between UPDATE$_\mathcal{H}$ and UPDATE$_\mathcal{F}$ in Section 4.

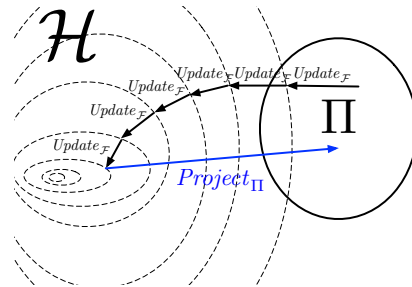

Figure 3: Depicting the PROPEL meta-algorithm.

**PROJECT$_\Pi$.** Projecting onto $\Pi$ can be implemented using program synthesis via imitation learning, i.e., by synthesizing a $\pi \in \Pi$ to best imitate demonstrations provided by a teaching oracle $h \in \mathcal{H}$. Recent work [58, 8, 60] has given practical heuristics for this task for various

---
**Algorithm 2** UPDATE$_\mathcal{F}$: neural policy gradient for mixed policies
---
1: **Input:** Neural Policy Class $\mathcal{F}$.     **Input:** Reference programmatic policy: $\pi$
2: **Input:** Step size: $\eta$.     **Input:** Regularization parameter: $\lambda$
3: Initialize neural policy: $f_0$     *//any standard randomized initialization*
4: **for** $j = 1, \ldots, m$ **do**
5:     $f_j \leftarrow f_{j-1} - \eta\lambda\nabla_\mathcal{F} J(\pi + \lambda f_{j-1})$     *//using DDPG [36], TRPO [47], etc., holding $\pi$ fixed*
6: **end for**
7: **Return:** $h \equiv \pi + \lambda f_m$
---

---
**Algorithm 3** PROJECT$_\Pi$: program synthesis via imitation learning
---
1: **Input:** Programmatic Policy Class: $\Pi$.     **Input:** Oracle policy: $h$
2: Roll-out $h$ on environment, get trajectory: $\tau_0 = (s^0, h(s^0), s^1, h(s^1), \ldots)$
3: Create supervised demonstration set: $\Gamma_0 = \{(s, h(s))\}$ from $\tau_0$
4: Derive $\pi_0$ from $\Gamma_0$ via program synthesis     *//e.g., using methods in [58, 8]*
5: **for** $k = 1, \ldots, M$ **do**
6:     Roll-out $\pi_{k-1}$, creating trajectory: $\tau_k$
7:     Collect demonstration data: $\Gamma' = \{(s, h(s)) | s \in \tau_k\}$
8:     $\Gamma_k \leftarrow \Gamma' \cup \Gamma_{k-1}$     *//DAgger-style imitation learning [46]*
9:     Derive $\pi_k$ from $\Gamma_k$ via program synthesis     *//e.g., using methods in [58, 8]*
10: **end for**
11: **Return:** $\pi_M$
---

programmatic policy classes. Algorithm 3 shows one instantiation of PROJECT$_\Pi$ (based on DAgger [46]). One complication that arises is that finite-sample runs of such imitation learning approaches only return approximate solutions and so the projection is not exact. We characterize the impact of approximate projections in Section 4.

**Practical Considerations.** In practice, we often employ multiple gradient steps before taking a projection step (as also described in Algorithm 2), because the step size of individual (stochastic) gradient updates can be quite small. Another issue that arises in virtually all policy gradient approaches is that the gradient estimates can have very high variance [55, 33, 30]. We utilize low-variance policy gradient updates by using the reference $\pi$ as a proximal regularizer in function space [19].

For the projection step (Algorithm 3), in practice we often retain all previous roll-outs $\tau$ from all previous projection steps. It is straightforward to query the current oracle $h$ to provide demonstrations on the states $s \in \tau$ from previous roll-outs, which can lead to substantial savings in sample complexity with regards to executing roll-outs on the environment, while not harming convergence.

## 4   Theoretical Analysis

We start by viewing PROPEL through the lens of online learning in function space, independent of the specific parametric representation. This start point yields a convergence analysis of Alg. 1 in Section 4.1 under generic approximation errors. We then analyze the issues of policy class representation in Sections 4.2 and 4.3, and connect Algorithms 2 and 3 with the overall performance, under some simplifying conditions. In particular, Section 4.3 characterizes the update error in a possibly non-differentiable setting; to our knowledge, this is the first such analysis of its kind for reinforcement learning.

**Preliminaries.** We consider $\Pi$ and $\mathcal{F}$ to be subspaces of an ambient policy space $\mathcal{U}$, which is a vector space equipped with inner product $\langle \cdot, \cdot \rangle$, induced norm $\|u\| = \sqrt{\langle u, u \rangle}$, dual norm $\|v\|_* = \sup\{\langle v, u \rangle | \|u\| \leq 1\}$, and standard scaling & addition: $(au + bv)(s) = au(s) + bv(s)$ for $a, b \in \mathbb{R}$ and $u, v \in \mathcal{U}$. The cost functional of a policy $u$ is $J(u) = \int_\mathcal{S} c(s, u(s)) d\mu^u(s)$, where $\mu^u$ is the distribution of states induced by $u$. The joint policy class is $\mathcal{H} = \Pi \oplus \mathcal{F}$, by $\mathcal{H} = \{\pi + f | \forall \pi \in \Pi, f \in \mathcal{F}\}$.[2] Note that $\mathcal{H}$ is a subspace of $\mathcal{U}$, and inherits its vector space properties. Without affecting the analysis, we simply equate $\mathcal{U} \equiv \mathcal{H}$ for the remainder of the paper.

We assume that $J$ is convex in $\mathcal{H}$, which implies that subgradient $\partial J(h)$ exists (with respect to $\mathcal{H}$) [9]. Where $J$ is differentiable, we utilize the notion of a Fréchet gradient. Recall that a bounded linear operator $\nabla : \mathcal{H} \mapsto \mathcal{H}$ is called a Fréchet functional gradient of $J$ at $h \in \mathcal{H}$ if

$\lim_{\|g\|\to 0} \frac{J(h+g)-J(h)-\langle\nabla J(h),g\rangle}{\|g\|} = 0$. By default, $\nabla$ (or $\nabla_{\mathcal{H}}$ for emphasis) denotes the gradient with respect to $\mathcal{H}$, whereas $\nabla_{\mathcal{F}}$ defines the gradient in the restricted subspace $\mathcal{F}$.

## 4.1 PROPEL as (Approximate) Functional Mirror Descent

For our analysis, PROPEL can be viewed as approximating mirror descent in (infinite-dimensional) function space over a convex set $\Pi \subset \mathcal{H}$.[3] Similar to the finite-dimensional setting [40], we choose a strongly convex and smooth functional regularizer $R$ to be the mirror map. From the approximate mirror descent perspective, for each iteration $t$:

1. Obtain a noisy gradient estimate: $\widehat{\nabla}_{t-1} \approx \nabla J(\pi_{t-1})$
2. UPDATE$_{\mathcal{H}}(\pi)$ in $\mathcal{H}$ space: $\nabla R(h_t) = \nabla R(\pi_{t-1}) - \eta\widehat{\nabla}_{t-1}$ *(Note* UPDATE$_{\mathcal{H}} \neq$ UPDATE$_{\mathcal{F}}$)
3. Obtain approximate projection: $\pi_t = \text{PROJECT}_{\Pi}^R(h_t) \approx \text{argmin}_{\pi\in\Pi} D_R(\pi, h_t)$

$D_R(u,v) = R(u) - R(v) - \langle\nabla R(u), u-v\rangle$ is a Bregman divergence. Taking $R(h) = \frac{1}{2}\|h\|^2$ will recover projected functional gradient descent in $L2$-space. Here UPDATE becomes $h_t = \pi_{t-1} - \eta\widehat{\nabla}J(\pi_{t-1})$, and PROJECT solves for $\text{argmin}_{\pi\in\Pi}\|\pi - h_t\|^2$. While we mainly focus on this choice of $R$ in our experiments, note that other selections of $R$ lead to different UPDATE and PROJECT operators (e.g., minimizing KL divergence if $R$ is negative entropy).

The functional mirror descent scheme above may encounter two additional sources of error compared to standard mirror descent [40]. First, in the stochastic setting (also called bandit feedback [28]), the gradient estimate $\widehat{\nabla}_t$ may be biased, in addition to having high variance. One potential source of bias is the gap between UPDATE$_{\mathcal{H}}$ and UPDATE$_{\mathcal{F}}$. Second, the PROJECT step may be inexact. We start by analyzing the behavior of PROPEL under generic bias, variance, and projection errors, before discussing the implications of approximating UPDATE$_{\mathcal{H}}$ and PROJECT$_{\Pi}$ by Algs. 2 & 3, respectively. Let the bias be bounded by $\beta$, i.e., $\left\|\mathbb{E}[\widehat{\nabla}_t|\pi_t] - \nabla J(\pi_t)\right\|_* \leq \beta$ almost surely. Similarly let the variance of the gradient estimate be bounded by $\sigma^2$, and the projection error norm $\|\pi_t - \pi_t^*\| \leq \epsilon$. We state the expected regret bound below; more details and a proof appear in Appendix A.2.

**Theorem 4.1** (Expected regret bound under gradient estimation and projection errors). *Let $\pi_1, \ldots, \pi_T$ be a sequence of programmatic policies returned by Algorithm 1, and $\pi^*$ be the optimal programmatic policy. Choosing learning rate $\eta = \sqrt{\frac{1}{\sigma^2}(\frac{1}{T} + \epsilon)}$, we have the expected regret over $T$ iterations:*

$$\mathbb{E}\left[\frac{1}{T}\sum_{t=1}^{T} J(\pi_t)\right] - J(\pi^*) = O\left(\sigma\sqrt{\frac{1}{T} + \epsilon} + \beta\right). \tag{2}$$

The result shows that error $\epsilon$ from PROJECT and the bias $\beta$ do not accumulate and simply contribute an additive term on the expected regret.[4] The effect of variance of gradient estimate decreases at a $\sqrt{1/T}$ rate. Note that this regret bound is agnostic to the specific UPDATE and PROJECT operations, and can be applied more generically beyond the specific algorithmic choices used in our paper.

## 4.2 Finite-Sample Analysis under Vanilla Policy Gradient Update and DAgger Projection

Next, we show how certain instantiations of UPDATE and PROJECT affect the magnitude of errors and influence end-to-end learning performance from finite samples, under some simplifying assumptions on the UPDATE step. For this analysis, we simplify Alg. 2 into the case UPDATE$_{\mathcal{F}} \equiv$ UPDATE$_{\mathcal{H}}$. In particular, we assume programmatic policies in $\Pi$ to be parameterized by a vector $\theta \in \mathbb{R}^k$, and $\pi$ is differentiable in $\theta$ (e.g., we can view $\Pi \subset \mathcal{F}$ where $\mathcal{F}$ is parameterized in $\mathbb{R}^k$). We further assume the trajectory roll-out is performed in an exploratory manner, where action is taken uniformly random over finite set of $A$ actions, thus enabling the bound on the bias of gradient estimates via Bernstein's inequality. The PROJECT step is consistent with Alg. 3, i.e., using DAgger [45] under convex imitation loss, such as $\ell_2$ loss. We have the following high-probability guarantee:

**Theorem 4.2** (Finite-sample guarantee). *At each iteration, we perform vanilla policy gradient estimate of $\pi$ (over $\mathcal{H}$) using $m$ trajectories and, use DAgger algorithm to collect $M$ roll-outs for the*

rounds of the algorithm, we have that:*

$$\frac{1}{T}\sum_{t=1}^{T} J(\pi_t) - J(\pi^*) \le O\left(\sigma\sqrt{\frac{1}{T} + \frac{H}{M} + \sqrt{\frac{\log(T/\delta)}{M}}}\right) + O\left(\sigma\sqrt{\frac{\log(Tk/\delta)}{m}} + \frac{AH\log(Tk/\delta)}{m}\right)$$

*holds with probability at least $1 - \delta$, with $H$ being the task horizon, $A$ the cardinality of action space,
$\sigma^2$ the variance of policy gradient estimates, and $k$ the dimension $\Pi$'s parameterization.*

The expanded result and proof are included in Appendix A.3. The proof leverages previous analysis
from DAgger [46] and the finite sample analysis of vanilla policy gradient algorithm [32]. The
finite-sample regret bound scales linearly with the standard deviation $\sigma$ of the gradient estimate, while
the bias, which is the very last component of the RHS, scales linearly with the task horizon $H$. Note
that the standard deviation $\sigma$ can be exponential in task horizon $H$ in the worst case [32], and so it is
important to have practical implementation strategies to reduce the variance of the UPDATE operation.
While conducted in a stylized setting, this analysis provides insight in the relative trade-offs of
spending effort in obtaining more accurate projections versus more reliable gradient estimates.

### 4.3    Closing the gap between UPDATE$_\mathcal{H}$ and UPDATE$_\mathcal{F}$

Our functional mirror descent analysis rests on taking gradients in $\mathcal{H}$: UPDATE$_\mathcal{H}(\pi)$ involves
estimating $\nabla_\mathcal{H} J(\pi)$ in the $\mathcal{H}$ space. On the other hand, Algorithm 2 performs UPDATE$_\mathcal{F}(\pi)$ only in
the neural policy space $\mathcal{F}$. In either case, although $J(\pi)$ may be differentiable in the non-parametric
ambient policy space, it may not be possible to obtain a differentiable parametric programmatic
representation in $\Pi$. In this section, we discuss theoretical motivations to addressing a practical issue:
*How do we define and approximate the gradient $\nabla_\mathcal{H} J(\pi)$ under a parametric representation?* To our
knowledge, we are the first to consider such a theoretical question for reinforcement learning.

**Defining a consistent approximation of $\nabla_\mathcal{H} J(\pi)$.** The idea in UPDATE$_\mathcal{F}(\pi)$ (Line 8 of Alg. 1) is
to approximate $\nabla_\mathcal{H} J(\pi)$ by $\nabla_\mathcal{F} J(f)$, which has a differentiable representation, at some $f$ close to $\pi$
(under the norm). Under appropriate conditions on $\mathcal{F}$, we show that this approximation is valid.

**Proposition 4.3.** *Assume that (i) $J$ is Fréchet differentiable on $\mathcal{H}$, (ii) $J$ is also differentiable on
the restricted subspace $\mathcal{F}$, and (iii) $\mathcal{F}$ is dense in $\mathcal{H}$ (i.e., the closure $\overline{\mathcal{F}} = \mathcal{H}$). Then for any
fixed policy $\pi \in \Pi$, define a sequence of policies $f_k \in \mathcal{F}$, $k = 1, 2, \ldots$), that converges to $\pi$:
$\lim_{k\to\infty}\|f_k - \pi\| = 0$. We then have $\lim_{k\to\infty}\|\nabla_\mathcal{F} J(f_k) - \nabla_\mathcal{H} J(\pi)\|_* = 0$.*

Since the Fréchet gradient is unique in the ambient space $\mathcal{H}$, $\forall k$ we have $\nabla_\mathcal{H} J(f_k) = \nabla_\mathcal{F} J(f_k) \to
\nabla_\mathcal{H} J(\pi)$ as $k \to \infty$ (by Proposition 4.3). We thus have an asymptotically unbiased approximation of
$\nabla_\mathcal{H} J(\pi)$ via differentiable space $\mathcal{F}$ as: $\nabla_\mathcal{F} J(\pi) \triangleq \nabla_\mathcal{H} J(\pi) \triangleq \lim_{k\to\infty}\nabla_\mathcal{F} J(f_k)$.[5] Connecting to
the result from Theorem 4.1, let $\sigma^2$ be an upper bound on the policy gradient estimates in the *neural
policy class $\mathcal{F}$*, under an asymptotically unbiased approximation of $\nabla_\mathcal{H} J(\pi)$, the expected regret
bound becomes $\mathbb{E}\left[\frac{1}{T}\sum_{t=1}^{T} J(\pi_t)\right] - J(\pi^*) = O\left(\sigma\sqrt{\frac{1}{T} + \epsilon}\right)$.

**Bias-variance considerations of UPDATE$_\mathcal{F}(\pi)$** To further theoretically motivate a practical strategy
for UPDATE$_\mathcal{F}(\pi)$ in Algorithm 2, we utilize an equivalent proximal perspective of mirror descent
[10], where UPDATE$_\mathcal{H}(\pi)$ is equivalent to solving for $h' = \operatorname{argmin}_{h\in\mathcal{H}} \eta\langle\nabla_\mathcal{H} J(\pi), h\rangle + D_R(h, \pi)$.

**Proposition 4.4** (Minimizing a relaxed objective). *For a fixed programmatic policy $\pi$, with sufficiently
small constant $\lambda \in (0, 1)$, we have that*

$$\min_{h\in\mathcal{H}} \eta\langle\nabla_\mathcal{H} J(\pi), h\rangle + D_R(h, \pi) \le \min_{f\in\mathcal{F}} J(\pi + \lambda f) - J(\pi) + \langle\nabla J(\pi), \pi\rangle \qquad (3)$$

Thus, a relaxed UPDATE$_\mathcal{H}$ step is obtained by minimizing the RHS of (3), i.e., minimizing $J(\pi + \lambda f)$
over $f \in \mathcal{F}$. Each gradient descent update step is now $f' = f - \eta\lambda\nabla_\mathcal{F} J(\pi_t + \lambda f)$, corresponding
to Line 5 of Algorithm 2. For fixed $\pi$ and small $\lambda$, this relaxed optimization problem becomes
regularized policy optimization over $\mathcal{F}$, which is significantly easier. Functional regularization in
policy space around a fixed prior controller $\pi$ has demonstrated significant reduction in the variance

not exist under policy parameterization of $\Pi$.

of gradient estimate [19], at the expense of some bias. The below expected regret bound summarizes the impact of this increased bias and reduced variance, with details included in Appendix A.5.

**Proposition 4.5** (Bias-variance characterization of UPDATE$_\mathcal{F}$). *Assuming $J(h)$ is L-strongly smooth over $\mathcal{H}$, i.e., $\nabla_\mathcal{H} J(h)$ is L-Lipschitz continuous, approximating* UPDATE$_\mathcal{H}$ *by* UPDATE$_F$ *per Alg. 2 leads to the expected regret bound:* $\mathbb{E}\left[\frac{1}{T}\sum_{t=1}^{T} J(\pi_t)\right] - J(\pi^*) = O\left(\lambda\sigma\sqrt{\frac{1}{T}} + \epsilon + \lambda^2 L^2\right)$.

Compared to the idealized unbiased approximation in Proposition 4.3, the introduced bias here is related to the inherent smoothness property of cost functional $J(h)$ over the joint policy class $\mathcal{H}$, i.e., how close $J(\pi + \lambda f)$ is to its linear under-approximation $J(\pi) + \langle \nabla_\mathcal{H} J(\pi), \lambda f \rangle$ around $\pi$.

## 5  Experiments

We demonstrate the effectiveness of PROPEL in synthesizing programmatic controllers in three continuous control environments. For brevity and focus, this section primarily focuses on TORCS[6], a challenging race car simulator environment [59]. Empirical results on two additional classic control tasks, Mountain-Car and Pendulum, are provided in Appendix B; those results follow similar trends as the ones described for TORCS below, and further validate the convergence analysis of PROPEL.

**Experimental Setup.** We evaluate over five distinct tracks in the TORCS simulator. The difficulty of a track can be characterized by three properties; track length, track width, and number of turns. Our suite of tracks provides environments with varying levels of difficulty for the learning algorithm. The performance of a policy in the TORCS simulator is measured by the *lap time* achieved on the track. To calculate the lap time, the policies are allowed to complete a three-lap race, and we record the best lap time during this race. We perform the experiments with twenty-five random seeds and report the median lap time over these twenty-five trials. Some of the policies crash the car before completing a lap on certain tracks, even after training for 600 episodes. Such crashes are recorded as a lap

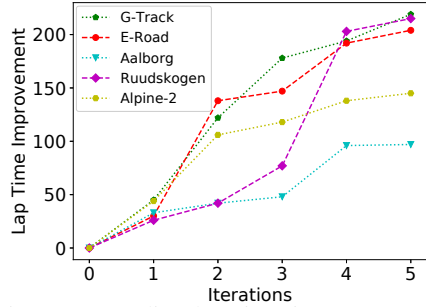

Figure 4: Median lap-time improvements during multiple iterations of PROPELPROG over 25 random seeds.

time of infinity while calculating the median. If the policy crashes for more than half the seeds, this is reported as CR in Tables 1 & 2. We choose to report the median because taking the crash timing as infinity, or an arbitrarily large constant, heavily skews other common measures such as the mean.

**Baselines.** Among recent state-of-the-art approaches to learning programmatic policies are NDPS [58] for high-level language policies, and VIPER [8] for learning tree-based policies. Both NDPS and VIPER rely on imitating a fixed (pre-trained) neural policy oracle, and can be viewed as degenerate versions of PROPEL that only run Lines 4-6 in Algorithm 1. We present two PROPEL analogues to NDPS and VIPER: (i) PROPELPROG: PROPEL using the high-level language of Figure 1 as the class of programmatic policies, similar to NDPS. (ii) PROPELTREE: PROPEL using regression trees, similar to VIPER. We also report results for PRIOR, which is a (sub-optimal) PID controller that is also used as the initial policy in PROPEL. In addition, to study generalization ability as well as safety behavior

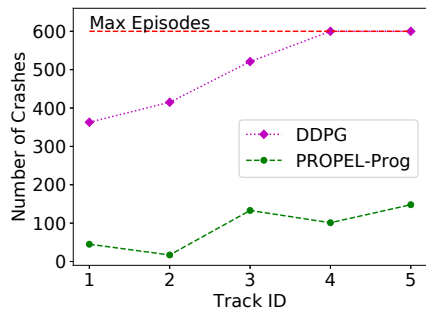

Figure 5: Median number of crashes during training of DDPG and PROPELPROG over 25 random seeds.

during training, we also include DDPG, a neural policy learned using the Deep Deterministic Policy Gradients [36] algorithm, with 600 episodes of training for each track. In principle, PROPEL and its analysis can accommodate different policy gradient subroutines. However, in the TORCS domain, other policy gradient algorithms such as PPO and TRPO failed to learn policies that are able to complete the considered tracks. We thus focus on DDPG as our main policy gradient component.

Table 1: *Performance results in* TORCS *over 25 random seeds. Each entry is formatted as Lap-time / Crash-ratio, reporting median lap time in seconds over all the seeds (lower is better) and ratio of seeds that result in crashes (lower is better). A lap time of* CR *indicates the agent crashed and could not complete a lap for more than half the seeds.*

| LENGTH | G-TRACK 3186M | E-ROAD 3260M | AALBORG 2588M | RUUDSKOGEN 3274M | ALPINE-2 3774M |
|---|---|---|---|---|---|
| PRIOR | 312.92 / 0.0 | 322.59 / 0.0 | 244.19 / 0.0 | 340.29 / 0.0 | 402.89 / 0.0 |
| DDPG | 78.82 / 0.24 | 89.71 / 0.28 | 101.06 / 0.40 | CR / 0.68 | CR / 0.92 |
| NDPS | 108.25 / 0.24 | 126.80 / 0.28 | 163.25 / 0.40 | CR / 0.68 | CR / 0.92 |
| VIPER | 83.60 / 0.24 | 87.53 / 0.28 | 110.57 / 0.40 | CR / 0.68 | CR / 0.92 |
| PROPELPROG | 93.67 / 0.04 | 119.17 / 0.04 | 147.28 / 0.12 | 124.58 / 0.16 | 256.59 / 0.16 |
| PROPELTREE | 78.33 / 0.04 | 79.39 / 0.04 | 109.83 / 0.16 | 118.80 / 0.24 | 236.01 / 0.36 |

Table 2: *Generalization results in* TORCS*, where rows are training and columns are testing tracks. Each entry is formatted as* PROPELPROG */ DDPG, and the number reported is the median lap time in seconds over all the seeds (lower is better).* CR *indicates the agent crashed and could not complete a lap for more than half the seeds.*

| | G-TRACK | E-ROAD | AALBORG | RUUDSKOGEN | ALPINE-2 |
|---|---|---|---|---|---|
| G-TRACK | - | 124 / CR | CR / CR | CR / CR | CR / CR |
| E-ROAD | 102 / 92 | - | CR / CR | CR / CR | CR / CR |
| AALBORG | 201 / 91 | 228 / CR | - | 217 / CR | CR / CR |
| RUUDSKOGEN | 131 / CR | 135 / CR | CR / CR | - | CR / CR |
| ALPINE-2 | 222 / CR | 231 / CR | 184 / CR | CR / CR | - |

**Evaluating Performance.** Table 1 shows the performance on the considered TORCS tracks. We see that PROPELPROG and PROPELTREE consistently outperform the NDPS [58] and VIPER [8] baselines, respectively. While DDPG outperforms PROPEL on some tracks, its volatility causes it to be unable to learn in some environments, and hence to crash the majority of the time. Figure 4 shows the consistent improvements made over the prior by PROPELPROG, over the iterations of the PROPEL algorithm. Appendix B contains similar results achieved on the two classic control tasks, MountainCar and Pendulum. Figure 5 shows that, compared to DDPG, our approach suffers far fewer crashes while training in TORCS.

**Evaluating Generalization.** To compare the ability of the controllers to perform on tracks not seen during training, we executed the learned policies on all the other tracks (Table 2). We observe that DDPG crashes significantly more often than PROPELPROG. This demonstrates the generalizability of the policies returned by PROPEL. Generalization results for the PROPELTREE policy are given in the appendix. In general, PROPELTREE policies are more generalizable than DDPG but less than PROPELPROG. On an absolute level, the generalization ability of PROPEL still leaves much room for improvement, which is an interesting direction for future work.

**Verifiability of Policies.** As shown in prior work [8, 58], parsimonious programmatic policies are more amenable to formal verification than neural policies. Unsurprisingly, the policies generated by PROPELTREE and PROPELPROG are easier to verify than DDPG policies. As a concrete example, we verified a smoothness property of the PROPELPROG policy using the Z3 SMT-solver [21] (more details in Appendix B). The verification terminated in $0.49$ seconds.

**Initialization.** In principle, PROPEL can be initialized with a random program, or a random policy trained using DDPG. In practice, the performance of PROPEL depends to a certain degree on the stability of the policy gradient procedure, which is DDPG in our experiments. Unfortunately, DDPG often exhibits high variance across trials and fares poorly in challenging RL domains. Specifically, in our TORCS experiments, DDPG fails on a number of tracks (similar phenomena have been reported in previous work that experiments on similar continuous control domains [30, 19, 58]). Agents obtained by initializing PROPEL with neural policies obtained via DDPG also fail on multiple tracks. Their performance over the five tracks is reported in Appendix B. In contrast, PROPEL can often finish the challenging tracks when initialized with a very simple hand-crafted programmatic prior.

# 6 Related Work

**Program Synthesis.** Program synthesis is the problem of automatically searching for a program within a language that fits a given specification [29]. Recent approaches to the problem have leveraged symbolic knowledge about program structure [27], satisfiability solvers [50, 31], and meta-learning techniques [39, 41, 22, 7] to generate interesting programs in many domains [3, 42, 4]. In most prior work, the specification is a logical constraint on the input/output behavior of the target program. However, there is also a growing body of work that considers program synthesis modulo optimality objectives [13, 15, 43], often motivated by machine learning tasks [39, 57, 26, 23, 58, 8, 60]. Synthesis of programs that imitates an oracle has been considered in both the logical [31] and the optimization [58, 8, 60] settings. The projection step in PROPEL builds on this prior work. While our current implementation of this step is entirely symbolic, in principle, the operation can also utilize contemporary techniques for learning policies that guide the synthesis process [39, 7, 49].

**Constrained Policy Learning.** Constrained policy learning has seen increased interest in recent years, largely due to the desire to impose side guarantees such as stability and safety on the policy's behavior. Broadly, there are two approaches to imposing constraints: specifying constraints as an additional cost function [1, 35], and explicitly encoding constraints into the policy class [2, 34, 19, 20, 12]. In some cases, these two approaches can be viewed as duals of each other. For instance, recent work that uses control-theoretic policies as a functional regularizer [34, 19] can be viewed from the perspective of both regularization (additional cost) and an explicitly constrained policy class (a specific mix of neural and control-theoretic policies). We build upon this perspective to develop the gradient update step in our approach.

**RL using Imitation Learning.** There are two ways to utilize imitation learning subroutines within RL. First, one can leverage limited-access or sub-optimal experts to speed up learning [44, 18, 14, 51]. Second, one can learn over two policy classes (or one policy and one model class) to achieve accelerated learning compared to using only one policy class [38, 17, 52, 16]. Our approach has some stylistic similarities to previous efforts [38, 52] that use a richer policy space to search for improvements before re-training the primary policy to imitate the richer policy. One key difference is that our primary policy is programmatic and potentially non-differentiable. A second key difference is that our theoretical framework takes a functional gradient descent perspective — it would be interesting to carefully compare with previous analysis techniques to find a unifying framework.

**RL with Mirror Descent.** The mirror descent framework has previously used to analyze and design RL algorithms. For example, Thomas et al. [56] and Mahadevan and Liu [37] use composite objective mirror descent, or COMID [25], which allows incorporating adaptive regularizers into gradient updates, thus offering connections to either natural gradient RL [56] or sparsity inducing RL algorithms [37]. Unlike in our work, these prior approaches perform projection into the same native, differentiable representation. Also, the analyses in these papers do not consider errors introduced by hybrid representations and approximate projection operators. However, one can potentially extend our approach with versions of mirror descent, e.g., COMID, that were considered in these efforts.

# 7 Conclusion and Future Work

We have presented PROPEL, a meta-algorithm based on mirror descent, program synthesis, and imitation learning, for programmatic reinforcement learning (PRL). We have presented theoretical convergence results for PROPEL, developing novel analyses to characterize approximate projections and biased gradients within the mirror descent framework. We also validated PROPEL empirically, and show that it can discover interpretable, verifiable, generalizable, performant policies and significantly outperform the state of the art in PRL.

The central idea of PROPEL is the use of imitation learning and combinatorial methods in implementing a projection operation for mirror descent, with the goal of optimization in a functional space that lacks gradients. While we have developed PROPEL in an RL setting, this idea is not restricted to RL or even sequential decision making. Future work will seek to exploit this insight in other machine learning and program synthesis settings.

**Acknowledgements.** This work was supported in part by United States Air Force Contract # FA8750-19-C-0092, NSF Award # 1645832, NSF Award # CCF-1704883, the Okawa Foundation, Raytheon, PIMCO, and Intel.

## Footnotes

[2]The operator $\oplus$ is not a direct sum, since $\Pi$ and $\mathcal{F}$ are not orthogonal.

[3]$\Pi$ can be convexified by considering *randomized* policies, as stochastic combinations of $\pi \in \Pi$ (cf. [35]).

[4]Other mirror descent-style analyses, such as in [52], lead to accumulation of errors over the rounds of learning $T$. One key difference is that we are leveraging the assumption of convexity of $J$ in the (infinite-dimensional) function space representation.

*imitation learning projection. Setting the learning rate $\eta = \sqrt{\frac{1}{\sigma^2}\left(\frac{1}{T} + \frac{H}{M} + \sqrt{\frac{\log(T/\delta)}{M}}\right)}$, after $T$

[5]We do not assume $J(\pi)$ to be differentiable when restricting to the policy subspace $\Pi$, i.e., $\nabla_\Pi J(\pi)$ may

[6]The code for the TORCS experiments can be found at: https://bitbucket.org/averma8053/propel

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
