[Supplementary Material]

# A  Theoretical Analysis

## A.1  Preliminaries and Notations

We formally define an ambient control policy space $\mathcal{U}$ to be a vector space equipped with inner product $\langle \cdot, \cdot \rangle : \mathcal{U} \times \mathcal{U} \mapsto \mathbb{R}$, which induces a norm $\|u\| = \sqrt{\langle u, u \rangle}$, and its dual norm defined as $\|v\|_* = \sup\{\langle v, u \rangle | \|u\| \leq 1\}$. While multiple ways to define the inner product exist, for concreteness we can think of the example of square-integrable stationary policies with $\langle u, v \rangle = \int_{\mathcal{S}} u(s)v(s)ds$. The addition operator $+$ between two policies $u, v \in \mathcal{U}$ is defined as $(u + v)(s) = u(s) + v(s)$ for all state $s \in \mathcal{S}$. Scaling $\lambda u + \kappa v$ is defined similarly for scalar $\lambda, \kappa$.

The cost functional of a control policy $u$ is defined as $J(u) = \int_0^\infty c(s(\tau), u(\tau))d\tau$, or $J(u) = \int_{\mathcal{S}} c(s, u(s))d\mu^u(s)$, where $\mu^u$ is the distribution of states induced by policy $u$. This latter example is equivalent to the standard notion of value function in reinforcement learning.

Separate from the parametric representation issues, both programmatic policy class $\Pi$ and neural policy class $\mathcal{F}$, and by extension - the joint policy class $\mathcal{H}$, are considered to live in the ambient vector space $\mathcal{U}$. We thus have a common and well-defined notion of distance between policies from different classes.

We make an important distinction between differentiability of $J(h)$ in the ambient policy space (non-parametric), versus differentiability in parameterization (parametric). For example, if $\Pi$ is a class of decision-tree based policy, policies in $\Pi$ may not be differentiable under representation. However, policies $\pi \in \Pi$ might still be differentiable when considered as points in the ambient vector space $\mathcal{U}$.

We will use the following standard notion of gradient and differentiability from functional analysis:

**Definition A.1** (Subgradients). The subgradient of $J$ at $h$, denoted $\partial J(h)$, is the non-empty set $\{g \in \mathcal{H} | \forall j \in \mathcal{H} : \langle j - h, g \rangle + J(h) \leq J(j)\}$

**Definition A.2** (Fréchet gradient). A bounded linear operator $\nabla : \mathcal{H} \mapsto \mathcal{H}$ is called Fréchet functional gradient of $J$ at $h \in \mathcal{H}$ if $\lim_{\|g\| \to 0} \frac{J(h+g) - J(h) - \langle \nabla J(h), g \rangle}{\|g\|} = 0$

The notions of convexity, smoothness and Bregman divergence are analogous to finite-dimensional setting:

**Definition A.3** (Strong convexity). A differentiable function $R$ is $\alpha-$strongly convex w.r.t norm $\|\cdot\|$ if $R(y) \geq R(x) + \langle \nabla R(x), y - x \rangle + \frac{\alpha}{2}\|y - x\|^2$

**Definition A.4** (Lipschitz continuous gradient smoothness). A differentiable function $R$ is $L_R-$strongly smooth w.r.t norm $\|\cdot\|$ if $\|\nabla R(x) - \nabla R(y)\|_* \leq L_R \|x - y\|$

**Definition A.5** (Bregman Divergence). For a strongly convex regularizer $R$, $D_R(x, y) = R(x) - R(y) - \langle \nabla R(y), x - y \rangle$ is the Bregman divergence between $x$ and $y$ (not necessarily symmetric)

The following standard result for Bregman divergence will be useful:

**Lemma A.1.** *[10] For all $x, y, z$ we have the identity $\langle \nabla R(x) - \nabla R(y), x - z \rangle = D_R(x, y) + D_R(z, x) - D_R(z, y)$. Since Bregman divergence is non-negative, a consequence of this identity is that $D_R(z, x) - D_R(z, y) \leq \langle \nabla R(x) - \nabla R(y), z - x \rangle$*

## A.2  Expected Regret Bound under Noisy Policy Gradient Estimates and Projection Errors

In this section, we show regret bound for the performance of the sequence of returned programs $\pi_1, \ldots, \pi_T$ of the algorithm. The analysis here is agnostic to the particular implementation of algorithm 2 and algorithm 3.

Let $R$ be a $\alpha-$strongly convex and $L_R-$smooth functional with respect to norm $\|\cdot\|$ on $\mathcal{H}$. The steps from algorithm 1 can be described as follows.

- Initialize $\pi_0 \in \Pi$. For each iteration $t$:
    1. Obtain a noisy estimate of the gradient $\widehat{\nabla} J(\pi_{t-1}) \approx \nabla J(\pi_{t-1})$
    2. Update in the $\mathcal{H}$ space: $\nabla R(h_t) = \nabla R(\pi_{t-1}) - \eta \widehat{\nabla} J(\pi_{t-1})$
    3. Obtain approximate projection $\pi_t = \text{PROJECT}_\pi^R(h_t) \approx \text{argmin}_{pi \in \Pi} D_R(\pi, h_t)$

This procedure is an approximate functional mirror descent scheme under bandit feedback. We will develop the following result, which is a more detailed version of 4.1 in the main paper.

In the statement below, $D$ is the diameter on $\Pi$ with respect to defined norm $\|\cdot\|$ (i.e., $D = \sup \|\pi - \pi'\|$). $L_J$ is the Lipschitz constant of the functional $J$ on $\mathcal{H}$. $\beta, \sigma^2$ are the bound on the bias and variance of the gradient estimate at each iteration, respectively. $\alpha$ and $_R$ are the strongly convex and smooth coefficients of the functional regularizer $R$. Finally, $\epsilon$ is the bound on the projection error with respect to the same norm $\|\cdot\|$.

**Theorem A.2** (Regret bound of returned policies). *Let $\pi_1, \ldots, \pi_T$ be a sequence of programmatic policies returned by algorithm 1 and $\pi^*$ be the optimal programmatic policy. We have the expected regret bound:*

$$\mathbb{E}\left[\frac{1}{T}\sum_{t=1}^{T} J(\pi_t)\right] - J(\pi^*) \leq \frac{L_R D^2}{\eta T} + \frac{\epsilon L_R D}{\eta} + \frac{\eta(\sigma^2 + L_J^2)}{\alpha} + \beta D$$

*In particular, choosing the learning rate $\eta = \sqrt{\frac{\frac{1}{T} + \epsilon}{\sigma^2}}$, the expected regret is simplified into:*

$$ \tag{4}$$

$$\mathbb{E}\left[\frac{1}{T}\sum_{t=1}^{T} J(\pi_t)\right] - J(\pi^*) = O\left(\sigma\sqrt{\frac{1}{T} + \epsilon} + \beta\right)$$

*Proof.* At each round $t$, let $\overline{\nabla}_t = \mathbf{E}[\widehat{\nabla}_t | \pi_t]$ be the conditional expectation of the gradient estimate. We will use the shorthand notation $\nabla_t = \nabla J(\pi_t)$. Denote the upper-bound on the bias of the estimate by $\beta_t$, i.e., $\left\|\overline{\nabla}_t - \nabla_t\right\|_* \leq \beta_t$ almost surely. Denote the noise of the gradient estimate by $\xi_t = \overline{\nabla}_t - \widehat{\nabla}_t$, and $\sigma_t^2 = \mathbf{E}\left[\left\|\widehat{\nabla}_t - \overline{\nabla}_t\right\|_*^2\right]$ is the variance of gradient estimate $\widehat{\nabla}_t$.

The projection operator is $\epsilon-$approximate in the sense that $\left\|\pi_t - \text{PROJECT}_\Pi^R(f_t)\right\| = \left\|\widehat{\text{PROJECT}}_\Pi^R(h_t) - \text{PROJECT}_\Pi^R(h_t)\right\| \leq \epsilon$ with some constant $\epsilon$, which reflects the statistical error of the imitation learning procedure. This projection error in general is independent of the choice of function classes $\Pi$ and $\mathcal{F}$. We will use the shorthand notation $\pi_t^* = \text{PROJECT}_\Pi^R(f_t)$ for the true Bregman projection of $h_t$ onto $\Pi$.

Due to convexity of $J$ over the space $\mathcal{H}$ (which includes $\Pi$), we have for all $\pi \in \Pi$:
$$J(\pi_t) - J(\pi) \leq \langle \nabla_t, \pi_t - \pi \rangle$$
We proceed to bound the RHS, starting with bounding the inner product where the actual gradient is replaced by the estimated gradient.

$$\langle \widehat{\nabla}_t, \pi_t - \pi \rangle = \frac{1}{\eta_t}\langle \nabla R(\pi_t) - \nabla R(h_{t+1}), \pi_t - \pi \rangle \tag{5}$$

$$= \frac{1}{\eta_t}\left(D_R(\pi, \pi_t) - D_R(\pi, h_{t+1}) + D_R(\pi_t, h_{t+1})\right) \tag{6}$$

$$\leq \frac{1}{\eta_t}\left(D_R(\pi, \pi_t) - D_R(\pi, \pi_{t+1}^*) - D_R(\pi_{t+1}^*, h_{t+1}) + D_R(\pi_t, h_{t+1})\right) \tag{7}$$

$$= \frac{1}{\eta_t}\Big(\underbrace{D_R(\pi, \pi_t) - D_R(\pi, \pi_{t+1})}_{\text{telescoping}} + \underbrace{D_R(\pi, \pi_{t+1}) - D_R(\pi, \pi_{t+1}^*)}_{\text{projection error}} \underbrace{- D_R(\pi_{t+1}^*, h_{t+1}) + D_R(\pi_t, h_{t+1})}_{\text{relative improvement}}\Big) \tag{8}$$

Equation (5) is due to the gradient update rule in $\mathcal{F}$ space. Equation (6) is derived from definition of Bregman divergence. Equation (7) is due to the generalized Pythagorean theorem of Bregman projection $D_R(x, y) \geq D_R(x, \text{PROJECT}_\Pi^R(x)) + D_R(\text{PROJECT}_\Pi^R(x), y)$. The RHS of equation (7) are decomposed into three components that will be bounded separately.

*Bounding projection error.* By lemma (A.1) we have

$$D_R(\pi, \pi_{t+1}) - D_R(\pi, \pi_{t+1}^*) \leq \langle \nabla R(\pi_{t+1}) - \nabla R(\pi_{t+1}^*), \pi - \pi_{t+1} \rangle \tag{9}$$

$$\leq \left\|\nabla R(\pi_{t+1}) - \nabla R(\pi_{t+1}^*)\right\| \|\pi - \pi_{t+1}\|_* \tag{10}$$

$$\leq L_R \left\| \pi_{t+1} - \pi_{t+1}^* \right\| D \leq \epsilon L_R D \tag{11}$$

Equation (10) is due to Cauchy–Schwarz. Equation (11) is due to Lipschitz smoothness of $\nabla R$ and definition of $\epsilon-$approximate projection.

*Bounding relative improvement.* This follows standard argument from analysis of mirror descent algorithm.

$$D_R(\pi_t, h_{t+1}) - D_R(\pi_{t+1}^*, h_{t+1}) = R(\pi_t) - R(\pi_{t+1}^*) + \langle \nabla R(h_{t+1}), \pi_{t+1}^* - \pi_t \rangle \tag{12}$$

$$\leq \langle \nabla R(\pi_t), \pi_t - \pi_{t+1}^* \rangle - \frac{\alpha}{2} \left\| \pi_{t+1}^* - \pi_t \right\|_*^2 + \langle \nabla R(h_{t+1}), \pi_{t+1}^* - \pi_t \rangle \tag{13}$$

$$= -\eta_t \langle \widehat{\nabla}_t, \pi_{t+1}^* - \pi_t \rangle - \frac{\alpha}{2} \left\| \pi_{t+1}^* - \pi_t \right\|^2 \tag{14}$$

$$\leq \frac{\eta_t^2}{2\alpha} \left\| \widehat{\nabla}_t \right\|_*^2 \leq \frac{\eta_t^2}{\alpha} (\sigma_t^2 + L_J^2) \tag{15}$$

Equation (13) is from the $\alpha-$strong convexity property of regularizer $R$. Equation (14) is by definition of the gradient update. Combining the bounds on the three components and taking expectation, we thus have

$$\mathbb{E}\left[ \langle \widehat{\nabla}_t, \pi_t - \pi \rangle \right] \leq \frac{1}{\eta_t} \left( D_R(\pi, \pi_t) - D_R(\pi, \pi_{t+1}) + \epsilon L_R D + \frac{\eta_t^2}{\alpha}(\sigma_t^2 + L_J^2) \right) \tag{16}$$

Next, the difference between estimated gradient $\widehat{\nabla}_t$ and actual gradient $\nabla_t$ factors into the bound via Cauchy-Schwarz:

$$\mathbb{E}\left[ \langle \nabla_t - \widehat{\nabla}_t, \pi_t - \pi \rangle \right] \leq \left\| \nabla_t - \mathbb{E}[\widehat{\nabla}_t] \right\|_* \|\pi_t - \pi\| \leq \beta_t D \tag{17}$$

The results can be deduced from equations (16) and (17).

**Unbiased gradient estimates.** For the case when the gradient estimate is unbiased, assume the variance of the noise of gradient estimates is bounded by $\sigma^2$, we have the expected regret bound for all $pi \in \Pi$

$$\mathbb{E}\left[ \frac{1}{T} \sum_{t=1}^{T} J(\pi_t) \right] - J(\pi) \leq \frac{L_R D^2}{\eta T} + \frac{\epsilon L_R D}{\eta} + \frac{\eta(\sigma^2 + L_J^2)}{\alpha} \tag{18}$$

here to clarify, $L_R$ is the smoothness coefficient of regularizer $R$ (i.e., the gradient of $R$ is $L_R$-Lipschitz, $L_J$ is Lipschitz constant of $J$, $D$ is the diameter of $\Pi$ under norm $\|\cdot\|$, $\sigma^2$ is the upper-bound on the variance of gradient estimates, and $\epsilon$ is the error from the projection procedure (i.e., imitation learning loss).

We can set learning rate $\eta = \sqrt{\frac{\frac{1}{T}+\epsilon}{\sigma^2}}$ to observe that the expected regret is bounded by $O(\sigma\sqrt{\frac{1}{T}+\epsilon})$.

**Biased gradient estimates.** Assume that the bias of gradient estimate at each round is upper-bounded by $\beta_t \leq \beta$. Similar to before, combining inequalities from (16) and (17), we have

$$\mathbb{E}\left[ \frac{1}{T} \sum_{t=1}^{T} J(\pi_t) \right] - J(\pi) \leq \frac{L_R D^2}{\eta T} + \frac{\epsilon L_R D}{\eta} + \frac{\eta(\sigma^2 + L_J^2)}{\alpha} + \beta D \tag{19}$$

Similar to before, we can set learning rate $\eta = \sqrt{\frac{\frac{1}{T}+\epsilon}{\sigma^2}}$ to observe that on the expected regret is bounded by $O(\sigma\sqrt{\frac{1}{T}+\epsilon}+\beta)$. Compared to the bound on (18), in the biased case, the extra regret incurred per bound is simply a constant, and does not depend on $T$. $\qquad\square$

## A.3 Finite-Sample Analysis

In this section, we provide overall finite-sample analysis for PROPEL under some simplifying assumptions. We first consider the case where exact gradient estimate is available, before extending the result to the general case of noisy policy gradient update. Combining the two steps will give us the proof for the following statement (theorem 4.2 in the main paper)

**Theorem A.3** (Finite-sample guarantee). *At each iteration, we perform vanilla policy gradient estimate of $\pi$ (over $\mathcal{H}$) using $m$ trajectories and use DAgger algorithm to collect $M$ roll-outs. Setting*

*the learning rate $\eta = \sqrt{\frac{1}{\sigma^2}\left(\frac{1}{T} + \frac{H}{M} + \sqrt{\frac{\log(T/\delta)}{M}}\right)}$, after $T$ rounds of the algorithm, we have that*

$$\frac{1}{T}\sum_{t=1}^{T} J(\pi_t) - J(\pi^*) \leq O\left(\sigma\sqrt{\frac{1}{T} + \frac{H}{M} + \sqrt{\frac{\log(T/\delta)}{M}}}\right) + O\left(\sigma\sqrt{\frac{\log(Tk/\delta)}{m}} + \frac{AH\log(Tk/\delta)}{m}\right)$$

*holds with probability at least $1 - \delta$, with $H$ the task horizon, $A$ the cardinality of action space, $\sigma^2$ the variance of policy gradient estimates, and $k$ the dimension $\Pi$'s parameterization.*

**Exact gradient estimate case.** Assuming that the policy gradients can be calculated exactly, it is straight-forward to provide high-probability guarantee for the effect of the projection error. We start with the following result, adapted from [45] for the case of projection error bound. In this version of DAgger, we assume that we only collect a single *(state, expert action)* pair from each trajectory roll-out. Result is similar, with tighter bound, when multiple data points are collected along the trajectory.

**Lemma A.4** (Projection error bound from imitation learning procedure). *Using DAgger as the imitation learning sub-routine for our* PROJECT *operator in algorithm 3, let $M$ be the number of trajectories rolled-out for learning, and $H$ be the horizon of the task. With probability at least $1 - \delta$, we have*

$$D_R(\pi, \pi^*) \leq \widetilde{O}(1/M) + \frac{2\ell_{max}(1 + H)}{M} + \sqrt{\frac{2\ell_{max}\log(1/\delta))}{M}}$$

*where $\pi$ is the result of* PROJECT*, $\pi^*$ is the true Bregman projection of $h$ onto $\Pi$, and $\ell_{max}$ is the maximum value of the imitation learning loss function $D_R(\cdot, \cdot)$*

The bound in lemma A.4 is simpler than previous imitation learning results with cost information ([44, 45]. The reason is that the goal of the PROJECT operator is more modest. Since we only care about the distance between the empirical projection $\pi$ and the true projection $\pi^*$, the loss objective in imitation learning is simplified (i.e., this is only a regret bound), and we can disregard how well policies in $\Pi$ can imitate the expert $h$, as well as the performance of $J(\pi)$ relative to the true cost from the environment $J(h)$.

A consequence of this lemma is that for the number of trajectories at each round of imitation learning $M = O(\frac{\log 1/\delta}{\epsilon^2}) + O(\frac{H}{\epsilon})$, we have $D_R(\pi_t, \pi_t^*) \leq \epsilon$ with probability at least $1 - \delta$. Applying union bound across $T$ rounds of learning, we obtain the following guarantee (under no gradient estimation error)

**Proposition A.5** (Finite-sample Projection Error Bound). *To simplify the presentation of the result, we consider $L_R, D, L, \alpha$ to be known constants. Using DAgger algorithm to collect $M = O(\frac{\log T/\delta}{\epsilon^2}) + O(\frac{H}{\epsilon})$ roll-outs at each iteration, we have the following regret guarantee after $T$ rounds of our main algorithm:*

$$\frac{1}{T}\sum_{t=1}^{T} J(\pi_t) - J(\pi^*) \leq O\left(\frac{1}{\eta T} + \frac{\epsilon}{\eta} + \eta\right)$$

*with probability at least $1 - \delta$. Consequently, setting $\eta = \sqrt{\frac{1}{T} + \frac{H}{M} + \sqrt{\frac{\log(T/\delta)}{M}}}$, we have that*

$$\frac{1}{T}\sum_{t=1}^{T} J(\pi_t) - J(\pi^*) \leq O\left(\sqrt{\frac{1}{T} + \frac{H}{M} + \sqrt{\frac{\log(T/\delta)}{M}}}\right)$$

*with probability at least $1 - \delta$*

Note that the dependence on the time horizon of the task is sub-linear. This is different from standard imitation learning regret bounds, which are often at least linear in the task horizon. The main reason is that our comparison benchmark $\pi^*$ does live in the space $\Pi$, whereas for DAgger, the expert policy may not reside in the same space.

**Noisy gradient estimate case.** We now turn to the issue of estimating the gradient of $\nabla J(\pi)$. We make the following simplifying assumption about the gradient estimation:

- The $\pi$ is parameterized by vector $\theta \in \mathbb{R}^k$ (such as a neural network). The parameterization is differentiable with respect to $\theta$ (Alternatively, we can view $\Pi$ as a differentiable subspace of $\mathcal{F}$, in which case we have $\mathcal{H} = \mathcal{F}$)

- At each UPDATE loop, the policy is rolled out $m$ times to collect the data, each trajectory has horizon length $H$

- For each visited state $s \sim d_h$, the policy takes a uniformly random action $a$. The action space is finite with cardinality $A$.

- The gradient $\nabla h_\theta$ is bounded by $B$

The gradient estimate is performed consistent with a generic policy gradient scheme, i.e.,

$$\widehat{\nabla} J(\theta) = \frac{A}{m} \sum_{i=1}^{H} \sum_{j=1}^{m} \nabla \pi_\theta(a_i^j | s_i^j, \theta) \widehat{Q}_i^j$$

where $\widehat{Q}_i^j$ is the estimated cost-to-go [55].

Taking uniform random exploratory actions ensures that the samples are i.i.d. We can thus apply Bernstein's inequality to obtain the bound between estimated gradient and the true gradient. Indeed, with probability at least $1 - \delta$, we have that the following bound on the bias component-wise:

$$\left\| \widehat{\nabla} J(\theta) - \nabla J(\theta) \right\|_\infty \leq \beta \text{ when } m \geq \frac{(2\sigma^2 + 2AHB\frac{\beta}{3}) \log \frac{k}{\delta}}{\beta^2}$$

which leads to similar bound with respect to $\|\cdot\|_*$ (here we leverage the equivalence of norms in finite dimensional setting):

$$\left\| \nabla_t - \widehat{\nabla}_t \right\|_* \leq \beta \text{ when } m = O\left( \frac{(\sigma^2 + AHB\beta) \log \frac{k}{\delta}}{\beta^2} \right)$$

Applying union bound of this result over $T$ rounds of learning, and combining with the result from proposition (A.5), we have the following finite-sample guarantee in the simplifying policy gradient update. This is also the more detailed statement of theorem 4.2 in the main paper.

**Proposition A.6** (Finite-sample Guarantee under Noisy Gradient Updates and Projection Error). *At each iteration, we perform policy gradient estimate using* $m = O(\frac{(\sigma^2 + AHB\beta) \log \frac{Tk}{\delta}}{\beta^2})$ *trajectories and use DAgger algorithm to collect* $M = O(\frac{\log T/\delta}{\epsilon^2}) + O(\frac{H}{\epsilon})$ *roll-outs. Setting the learning rate* $\eta = \sqrt{\frac{1}{\sigma^2} \left( \frac{1}{T} + \frac{H}{M} + \sqrt{\frac{\log(T/\delta)}{M}} \right)}$, *after* $T$ *rounds of the algorithm, we have that*

$$\frac{1}{T} \sum_{t=1}^{T} J(\pi_t) - J(\pi^*) \leq O\left( \sigma \sqrt{\frac{1}{T} + \frac{H}{M} + \sqrt{\frac{\log(T/\delta)}{M}}} \right) + \beta$$

*with probability at least* $1 - \delta$.

*Consequently, we also have the following regret bound:*

$$\frac{1}{T} \sum_{t=1}^{T} J(\pi_t) - J(\pi^*) \leq O\left( \sigma \sqrt{\frac{1}{T} + \frac{H}{M} + \sqrt{\frac{\log(T/\delta)}{M}}} \right) + O\left( \sigma \sqrt{\frac{\log(Tk/\delta)}{m}} + \frac{AH \log(Tk/\delta)}{m} \right)$$

*holds with probability at least* $1 - \delta$, *where again* $H$ *is the task horizon,* $A$ *is the cardinality of action space, and* $k$ *is the dimension of function class* $\Pi$'s *parameterization.*

*Proof.* (For both proposition (A.6) and (A.5)). The results follow by taking the inequality from equation (19), and by solving for $\epsilon$ and $\beta$ explicitly in terms of relevant quantities. Based on the specification of $M$ and $m$, we obtain the necessary precision for each round of learning in terms of number of trajectories:

$$\beta = O(\sigma \frac{\log(k/\delta)}{m} + \frac{AHB \log(k/\delta)}{m})$$

$$\epsilon = O(\frac{H}{M} + \sqrt{\frac{\log(1/\delta)}{M}})$$

Setting the learning rate $\eta = \sqrt{\frac{1}{\sigma^2}\left(\frac{1}{T} + \epsilon\right)}$ and rearranging the inequalities lead to the desired bounds. $\qquad\square$

The regret bound depends on the variance $\sigma^2$ of the policy gradient estimates. It is well-known that vanilla policy gradient updates suffer from high variance. We instead use functional regularization technique, based on CORE-RL, in the practical implementation of our algorithm. The CORE-RL subroutine has been demonstrated to reduce the variance in policy gradient updates [19].

## A.4 Defining a consistent approximation of $\nabla_{\mathcal{H}} J(\pi)$ - Proof of Proposition 4.3

We are using the notion of Fréchet derivative to define gradient of differentiable functional. Note that while Gateaux derivative can also be utilized, Fréchet derivative ensures continuity of the gradient operator that would be useful for our analysis.

**Definition A.6** (Fréchet gradient). A bounded linear operator $\nabla : \mathcal{H} \mapsto \mathcal{H}$ is called Fréchet functional gradient of $J$ at $h \in \mathcal{H}$ if $\lim_{\|g\| \to 0} \frac{J(h+g) - J(h) - \langle \nabla J(h), g \rangle}{\|g\|} = 0$

We make the following assumption about $\mathcal{H}$ and $\mathcal{F}$. One interpretation of this assumption is that the space of policies $\Pi$ and $\mathcal{F}$ that we consider have the property that a programmatic policy $\pi \in \Pi$ can be well-approximated by a large space of neural policies $f \in \mathcal{F}$.

**Assumption 1.** *$J$ is Fréchet differentiable on $\mathcal{H}$. $J$ is also differentiable on the restricted subspace $\mathcal{F}$. And $\mathcal{F}$ is dense in $\mathcal{H}$ (i.e., the closure $\overline{\mathcal{F}} = \mathcal{H}$)*

It is then clear that $\forall f \in \mathcal{F}$ the Fréchet gradient $\nabla_{\mathcal{F}} J(f)$, restricted to the subspace $\mathcal{F}$ is equal to the gradient of $f$ in the ambient space $\mathcal{H}$ (since Fréchet gradient is unique). In general, given $\pi \in \Pi$ and $f \in \mathcal{F}$, $\pi + f$ is not necessarily in $\mathcal{F}$. However, the restricted gradient on subspace $\mathcal{F}$ of $J(\pi + f)$ can be defined asymptotically.

**Proposition A.7.** *Fixing a policy $\pi \in \Pi$, define a sequence of policies $f_k \in \mathcal{F}$, $k = 1, 2, \ldots$ that converges to $\pi$: $\lim_{k \to \infty} \|f_k - g\| = 0$, we then have $\lim_{k \to \infty} \|\nabla_{\mathcal{F}} J(f_k) - \nabla_{\mathcal{H}} J(\pi)\|_* = 0$*

*Proof.* Since Fréchet derivative is a continuous linear operator, we have $\lim_{k \to \infty} \|\nabla_{\mathcal{H}} J(f_k) - \nabla_{\mathcal{H}} J(\pi)\|_* = 0$. By the reasoning above, for $f \in \mathcal{F}$, the gradient $\nabla_{\mathcal{F}} J(f)$ defined via restriction to the space $\mathcal{F}$ does not change compared to $\nabla_{\mathcal{H}} J(f)$, the gradient defined over the ambient space $\mathcal{H}$. Thus we also have $\lim_{k \to \infty} \|\nabla_{\mathcal{F}} J(f_k) - \nabla_{\mathcal{H}} J(\pi)\|_* = 0$. By the same argument, we also have that for any given $\pi \in \Pi$ and $f \in \mathcal{F}$, even if $\pi + f \notin \mathcal{F}$, the gradient $\nabla_{\mathcal{F}} J(\pi + f)$ with respect to the $\mathcal{F}$ can be approximated similarly. $\qquad\square$

Note that we are not assuming $J(\pi)$ to be differentiable when restricting to the policy subspace $\Pi$.

## A.5 Theoretical motivation for Algorithm 2 - Proof of Proposition 4.4 and 4.5

We consider the case where $\Pi$ is not differentiable by parameterization. Note that this does not preclude $J(\pi)$ for $\pi \in \Pi$ to be differentiable in the non-parametric function space. Two complications arise compared to our previous approximate mirror descent procedure. First, for each $\pi \in \Pi$, estimating the gradient $\nabla J(\pi)$ (which may not exist under certain parameterization, per section 4.3) can become much more difficult. Second, the update rule $\nabla R(\pi) - \nabla_{\mathcal{F}} J(\pi)$ may not be in the dual space of $\mathcal{F}$, as in the simple case where $\Pi \subset \mathcal{F}$, thus making direct gradient update in the $\mathcal{F}$ space inappropriate.

**Assumption 2.** *$J$ is convex in $\mathcal{H}$.*

By convexity of $J$ in $\mathcal{H}$, sub-gradients $\partial J(h)$ exists for all $h \in \mathcal{H}$. In particular, $\partial J(\pi)$ exists for all $\pi \in \Pi$. Note that $\partial J(\pi)$ reflects sub-gradient of $\pi$ with respect to the ambient policy space $\mathcal{H}$.

We will make use of the following equivalent perspective to mirror descent[10], which consists of two-step process for each iteration $t$

1. Solve for $h_{t+1} = \operatorname{argmin}_{h \in \mathcal{H}} \eta \langle \partial J(\pi_t), h \rangle + D_R(h, \pi_t)$
2. Solve for $\pi_{t+1} = \operatorname{argmin}_{\pi \in \Pi} D_R(\pi, h_{t+1})$

We will show how this version of the algorithm motivates our main algorithm. Consider step 1 of the main loop of PROPEL, where given a fixed $\pi \in \Pi$, the optimization problem within $\mathcal{H}$ is

$$(\text{OBJECTIVE\_1}) = \min_{h \in \mathcal{H}} \eta \langle \partial J(\pi), h \rangle + D_R(h, \pi) \tag{20}$$

Due to convexity of $\mathcal{H}$ and the objective, problem (OBJECTIVE\_1) is equivalent to:

$$(\text{OBJECTIVE\_1}) = \min \langle \partial J(\pi), h \rangle \tag{21}$$
$$\text{s.t. } D_R(h, \pi) \leq \tau \tag{22}$$

where $\tau$ depends on $\eta$. Since $\pi$ is fixed, this optimization problem can be relaxed by choosing $\lambda \in [0, 1]$, and a set of candidate policies $h = \pi + \lambda f$, for all $f \in \mathcal{F}$, such that $D_R(h, \pi) \leq \tau$ is satisfied (Selection of $\lambda$ is possible with bounded spaces). Since this constraint set is potentially a restricted set compared to the space of policies satisfying inequality (22), the optimization problem (20) is relaxed into:

$$(\text{OBJECTIVE\_1}) \leq (\text{OBJECTIVE\_2}) = \min_{f \in \mathcal{F}} \langle \partial J(\pi), \pi + \lambda f \rangle \tag{23}$$

Due to convexity property of $J$, we have

$$\langle \partial J(\pi), \lambda f \rangle = \langle \partial J(\pi), \pi + \lambda f - \pi \rangle \leq J(\pi + \lambda f) - J(\pi) \tag{24}$$

The original problem OBJECTIVE\_1 is thus upper bounded by:

$$\min_{h \in \mathcal{H}} \eta \langle \partial J(\pi), h) \rangle + D_R(h, \pi) \leq \min_{f \in \mathcal{F}} J(\pi + \lambda f) - J(\pi) + \langle \partial J(\pi), \pi \rangle$$

Thus, a relaxed version of original optimization problem OBJECTIVE\_1 can be obtained by minizi-iming $J(\pi + \lambda f)$ over $f \in \mathcal{F}$ (note that $\pi$ is fixed). This naturally motivates using functional regularization technique, such as CORE-RL algorithm [19], to update the parameters of differentiable function $f$ via policy gradient descent update:

$$f' = f - \eta \lambda \nabla_{\mathcal{F}} \lambda J(\pi + \lambda f)$$

where the gradient of $J$ is taken with respect to the parameters of $f$ (neural networks). This is exactly the update step in algorithm 2 (also similar to iterative updte of CORE-RL algorithm), where the neural network policy is regularized by a prior controller $\pi$.

**Statement and Proof of Proposition 4.5**

**Proposition A.8** (Regret bound for the relaxed optimization objective). *Assuming $J(h)$ is L-strongly smooth over $\mathcal{H}$, i.e., $\nabla_{\mathcal{H}} J(h)$ is L-Lipschitz continuous, approximating* UPDATE$_{\mathcal{H}}$ *by* UPDATE$_F$ *per Alg. 2 leads to the expected regret bound:* $\mathbb{E}\left[ \frac{1}{T} \sum_{t=1}^{T} J(\pi_t) \right] - J(\pi^*) = O\left( \lambda \sigma \sqrt{\frac{1}{T} + \epsilon} + \lambda^2 L^2 \right)$

*Proof.* Instead of focusing on the bias of the gradient estimate $\nabla_{\mathcal{H}} J(\pi)$, we will shift our focus on the alternative proximal formulation of mirror descent, under optimization and projection errors. In particular, at each iteration $t$, let $h_{t+1}^* = \text{argmin}_{h \in \mathcal{H}} \eta \langle \nabla J(\pi_t), h \rangle + D_R(h, \pi_t)$ and let the optimization error be defined as $\beta_t$ where $\nabla R(h_{t+1}) = \nabla R(h_{t+1}^*) + \beta_t$. Note here that this is different from (but related to) the notion of bias from gradient estimate of $\nabla J(\pi)$ used in theorem 4.1 and theorem A.2. The projection error from imitation learning procedure is defined similarly to theorem 4.1: $\pi_{t+1}^* = \text{argmin}_{\pi \in \Pi} D_R(\pi, h_{t+1})$ is the true projection, and $\left\| \pi_{t+1} - \pi_{t+1}^* \right\| \leq \epsilon$.

We start with similar bounding steps to the proof of theorem 4.1:

$$\langle \nabla J(\pi_t), \pi_t - \pi \rangle = \frac{1}{\eta} \langle \nabla R(h_{t+1}^*) - \nabla R(\pi_t), \pi_t - \pi \rangle$$

$$= \frac{1}{\eta} \left( \langle \nabla R(h_{t+1}) - \nabla R(\pi_t), \pi_t - \pi \rangle - \langle \beta_t, \pi_t - \pi \rangle \right)$$

$$= \frac{1}{\eta} \underbrace{\left( D_R(\pi, \pi_t) - D_R(\pi, h_{t+1}) + D_R(\pi_t, h_{t+1}) \right)}_{\text{component\_1}} + \underbrace{\frac{1}{\eta} \langle \beta_t, \pi_t - \pi \rangle}_{\text{component\_2}} \tag{25}$$

As seen from the proof of theorem A.2, component\_1 can be upperbounded by:
$\frac{1}{\eta} \left( \underbrace{D_R(\pi, \pi_t) - D_R(\pi, \pi_{t+1})}_{\text{telescoping}} + \underbrace{D_R(\pi, \pi_{t+1}) - D_R(\pi, \pi_{t+1}^*)}_{\text{projection error}} \underbrace{- D_R(\pi_{t+1}^*, h_{t+1}) + D_R(\pi_t, h_{t+1})}_{\text{relative improvement}} \right)$

The bound on projection error is identical to theorem A.2:

$$D_R(\pi, \pi_t) - D_R(\pi, \pi_{t+1}^*) \leq \epsilon L_R D \tag{26}$$

The bound on relative improvement is slightly different:

$$D_R(\pi_t, h_{t+1}) - D_R(\pi_{t+1}^*, h_{t+1}) = R(\pi_t) - R(\pi_{t+1}^*) + \langle \nabla R(h_{t+1}), \pi_{t+1}^* - \pi_t \rangle$$

$$= R(\pi_t) - R(\pi_{t+1}^* + \langle \nabla R(h_{t+1}^*), \pi_{t+1}^* - \pi_t \rangle) + \langle \beta_t, \pi_{t+1}^* - \pi_t \rangle$$

$$\leq \langle \nabla R(\pi_t), \pi_t - \pi_{t+1}^* \rangle - \frac{\alpha}{2} \left\| \pi_{t+1}^* - \pi_t \right\|^2 + \langle \nabla R(h_{t+1}^*), \pi_{t+1}^* - \pi_t \rangle + \langle \beta_t, \pi_{t+1}^* - \pi_t \rangle$$

$$= -\eta \langle \nabla J_{\mathcal{H}}(\pi_t), \pi_{t+1}^* - \pi_t \rangle - \frac{\alpha}{2} \left\| \pi_{t+1}^* - \pi_t \right\|^2 + \langle \beta_t, \pi_{t+1}^* - \pi_t \rangle \qquad (27)$$

$$\leq \frac{\eta^2}{2\alpha} \left\| \nabla_{\mathcal{H}} J(\pi_t) \right\|_*^2 + \langle \beta_t, \pi_{t+1}^* - \pi_t \rangle$$

$$\leq \frac{\eta^2}{2\alpha} L_J^2 + \langle \beta_t, \pi_{t+1}^* - \pi_t \rangle \qquad (28)$$

Note here that the gradient $\nabla_{\mathcal{H}} J(\pi_t)$ is not the result of estimation. Combining equations (25), (26), (27), (28), we have:

$$\langle \nabla J(\pi_t), \pi_t - \pi \rangle \leq \frac{1}{\eta} \Big( D_R(\pi, \pi_t) - D_R(\pi, \pi_{t+1}) + \epsilon L_R D + \frac{\eta^2}{2\alpha} L_J^2 + \langle \beta_t, \pi_{t+1}^* - \pi \rangle \Big) \quad (29)$$

Next, we want to bound $\beta_t$. Choose regularizer $R$ to be $\frac{1}{2} \left\| \cdot \right\|^2$ (consistent with the pseudocode in algorithm 2). We have that:

$$h_{t+1}^* = \operatorname*{argmin}_{h \in \mathcal{H}} \eta \langle \nabla J(\pi_t), h \rangle + \frac{1}{2} \left\| h - \pi_t \right\|^2$$

which is equivalent to:

$$h_{t+1}^* = \pi_t + \operatorname*{argmin}_{f \in \mathcal{F}} \eta \langle \nabla J(\pi_t), f \rangle + \frac{1}{2} \left\| f \right\|^2$$

Let $f_{t+1}^* = \operatorname{argmin}_{f \in \mathcal{F}} \eta \langle \nabla J(\pi_t), f \rangle + \frac{1}{2} \left\| f \right\|^2$. Taking the gradient over $f$, we can see that $f_{t+1}^* = -\eta \nabla J(\pi_t)$. Let $f_{t+1}$ be the minimizer of $\min_{f \in \mathcal{F}} J(\pi_t + \lambda f)$. We then have $h_{t+1}^* = \pi_t + f_{t+1}^*$ and $h_{t+1} = \pi + \lambda f_{t+1}$. Thus $\beta_t = h_{t+1} - h_{t+1}^* = f_{t+1} - f_{t+1}^*$.

On one hand, we have

$$J(\pi_t + \lambda f_{t+1}) \leq J(\pi_t + \omega f_{t+1}^*) \leq J(\pi_t) + \langle \nabla J(\pi_t), \omega f_{t+1}^* \rangle + \frac{L}{2} \left\| \omega f_{t+1}^* \right\|^2$$

due to optimality of $f_{t+1}$ and strong smoothness property of $J$. On the other hand, since $J$ is convex, we also have the first-order condition:

$$J(\pi_t + \lambda f_{t+1}) \geq J(\pi_t) + \langle \nabla J(\pi_t), \lambda f_{t+1} \rangle$$

Combine with the inequality above, and subtract $J(\pi_t)$ from both sides, and using the relationship $f_{t+1}^* = -\eta \nabla J(\pi_t)$, we have that:

$$\langle -\frac{1}{\eta} f_{t+1}^*, \lambda f_{t+1} \rangle \leq \langle -\frac{1}{\eta} f_{t+1}^*, \omega f_{t+1}^* \rangle + \frac{L\omega^2}{2} \left\| f_{t+1}^* \right\|^2$$

Since this is true $\forall \omega$, rearrange and choose $\omega$ such that $\frac{\omega}{\eta} - \frac{L\omega^2}{2} = -\frac{\lambda}{2\eta}$, namely $\omega = \frac{1 - \sqrt{1 - \lambda \eta L}}{L\eta}$, and complete the square, we can establish the bound that:

$$\left\| f_{t+1} - f_{t+1}^* \right\| \leq \eta (\lambda L)^2 B \qquad (30)$$

for $B$ the upperbound on $\left\| f_{t+1} \right\|$. We thus have $\left\| \beta_t \right\| = O(\eta (\lambda L)^2)$. Plugging the result from equation 30 into RHS of equation 29, we have:

$$\langle \nabla J(\pi_t), \pi_t - \pi \rangle \leq \frac{1}{\eta} \Big( D_R(\pi, \pi_t) - D_R(\pi, \pi_{t+1}) + \epsilon L_R D + \frac{\eta^2}{2\alpha} L_J^2 \Big) + \big( \eta (\lambda L)^2 B \big) \qquad (31)$$

Since $J$ is convex in $\mathcal{H}$, we have $J(\pi_t) - J(\pi) \leq \langle \nabla J(\pi_t), \pi_t - \pi \rangle$. Similar to theorem 4.1, setting $\eta = \sqrt{\frac{1}{\lambda^2 \sigma^2} (\frac{1}{T} + \epsilon)}$ and taking expectation on both sides, we have:

$$\mathbb{E} \left[ \frac{1}{T} \sum_{t=1}^{T} J(\pi_t) \right] - J(\pi^*) = O\Big( \lambda \sigma \sqrt{\frac{1}{T} + \epsilon} + \lambda^2 L^2 \Big) \qquad (32)$$

Note that unlike regret bound from theorem 4.1 under general bias, variance of gradient estimate and projection error, $\sigma^2$ here explicitly refers to the bound on neural-network based policy gradient

variance. The variance reduction of $\lambda\sigma$, at the expense of some bias, was also similarly noted in a recent functional regularization technique for policy gradient [19]. □

# B  Additional Experimental Results and Details

## B.1  TORCS

We generate controllers for cars in *The Open Racing Car Simulator* (TORCS) [59]. In its full generality TORCS provides a rich environment with input from up to 89 sensors, and optionally the 3D graphic from a chosen camera angle in the race. The controllers have to decide the values of 5 parameters during game play, which correspond to the acceleration, brake, clutch, gear and steering of the car.

Apart from the immediate challenge of driving the car on the track, controllers also have to make race-level strategy decisions, like making pit-stops for fuel. A lower level of complexity is provided in the Practice Mode setting of TORCS. In this mode all race-level strategies are removed. Currently, so far as we know, state-of-the-art DRL models are capable of racing only in Practice Mode, and this is also the environment that we use. Here we consider the input from 29 sensors, and decide values for the acceleration, steering, and braking actions.

We chose a suite of tracks that provide varying levels of difficulty for the learning algorithms. In particular, for the tracks Ruudskogen and Alpine-2, the DDPG agent is unable to reliably learn a policy that would complete a lap. We perform the experiments with twenty-five random seeds and report the median lap time over these twenty-five trials. However we note that the TORCS simulator is not deterministic even for a fixed random seed. Since we model the environment as a Markov Decision Process, this non-determinism is consistent with our problem statement.

For our Deep Reinforcement Learning agents we used standard open source implementations (with pre-tuned hyper-parameters) for the relevant domain.

All experiments were conducted on standard workstation with a 2.5 GHz Intel Core i7 CPU and a GTX 1080 Ti GPU card.

The code for the TORCS experiments can be found at: https://bitbucket.org/averma8053/propel

In Table 3 we show the lap time performance and crash ratios of PROPEL agents initialized with neural policies obtained via DDPG. As discussed in Section 5, DDPG often exhibits high variance across trials and this adversely affects the performance of the PROPEL agents when they are initialized via DDPG. In Table 4 we show generalization results for the PROPELTREE agent. As noted in Section 5, the generalization results for PROPELTREE are in between those of DDPG and PROPELPROG.

**Verified Smoothness Property.** For the program given in Figure 2 we proved using symbolic verification techniques, that $\forall k, \sum_{i=k}^{k+5} \|\mathbf{peek}(s[\texttt{RPM}], i+1) - \mathbf{peek}(s[\texttt{RPM}], i)\| < 0.003 \implies \|\mathbf{peek}(a[\texttt{Accel}], k+1) - \mathbf{peek}(a[\texttt{Accel}], k)\| < 0.63$. Here the function $\mathbf{peek}(., i)$ takes in a history/sequence of sensor or action values and returns the value at position $i$, . Intuitively, the above logical implication means that if the sum of the consecutive differences of the last six RPM sensor values is less than $0.003$, then the acceleration actions calculated at the last and penultimate step will not differ by more than $0.63$.

Table 3: *Performance results in* TORCS *of* PROPEL *agents initialized with neural policies obtained via* DDPG*, over 25 random seeds. Each entry is formatted as Lap-time / Crash-ratio, reporting median lap time in seconds over all the seeds (lower is better) and ratio of seeds that result in crashes (lower is better). A lap time of* CR *indicates the agent crashed and could not complete a lap for more than half the seeds.*

| | G-TRACK | E-ROAD | AALBORG | RUUDSKOGEN | ALPINE-2 |
| --- | --- | --- | --- | --- | --- |
| LENGTH | 3186M | 3260M | 2588M | 3274M | 3774M |
| PROPELPROG-DDPG | 97.76/.12 | 108.06/.08 | 140.48/.48 | CR / 0.68 | CR / 0.92 |
| PROPELTREE-DDPG | 78.47/0.16 | 85.46/.04 | CR / 0.56 | CR / 0.68 | CR / 0.92 |

Table 4: *Generalization results in* TORCS *for* PROPELTREE*, where rows are training and columns are testing tracks. Each entry is formatted as* PROPELPROG */ DDPG, and the number reported is the median lap time in seconds over all the seeds (lower is better).* CR *indicates the agent crashed and could not complete a lap for more than half the seeds.*

|  | G-TRACK | E-ROAD | AALBORG | RUUDSKOGEN | ALPINE-2 |
|---|---|---|---|---|---|
| G-TRACK | - | 95 | CR | CR | CR |
| E-ROAD | 84 | - | CR | CR | CR |
| AALBORG | 111 | CR | - | CR | CR |
| RUUDSKOGEN | 154 | CR | CR | - | CR |
| ALPINE-2 | CR | 276 | CR | CR | - |

Table 5: Performance results in Classic Control problems. Higher scores are better.

|  | MOUNTAINCAR | PENDULUM |
|---|---|---|
| PRIOR | 0.59 | -875.53 |
| DDPG | 96.35 | -135.83 |
| TRPO | 95.14 | -133.53 |
| NDPS | 68.34 | -440.82 |
| VIPER | 61.46 | -392.85 |
| PROPELPROG | 95.87 | -184.26 |
| PROPELTREE | 95.85 | -141.26 |

## B.2 Classic Control

We present results from two classic control problems, Mountain-Car (with continuous actions) and Pendulum, in Table 5. We use the OpenAI Gym implementations of these environments. More information about these environments can be found at the links: MountainCar and Pendulum.

In Mountain-Car the goal is to drive an under-powered car up the side of a mountain in as few time-steps as possible. In Pendulum, the goal is to swing a pendulum up so that it stays upright. In both the environments an episode terminates after a maximum of 200 time-steps.

In Table 5 we report the average score over 100 episodes for the listed agents, in both these environments. In Figure 6 and Figure 7 we show the improvements made over the prior by the PROPELPROG agent in MountainCar and Pendulum respectively, with each iteration of the PROPEL algorithm.

Figure 6: Score improvements in the MountainCar environment over iterations of PROPELPROG.

Figure 7: Score improvements in the Pendulum environment over iterations of PROPELPROG.