[Reviews · NeurIPS 2019]

Reviewer 1



The paper presents a very interesting idea: uses combinatorial methods only for projecting neural (or hybrid neural-symbolic) functions to the programmatic space. The projection operator uses program synthesis via imitation learning such as Dagger. The paper does not clearly explain why projection is much easier than searching in program space. A detailed explanation with an example will be needed. The evaluation results are very weak. It uses very simple RL tasks. The dropbox link https://www.dropbox.com/sh/qsgxk76xg9t5ow3/AACyJIkqwhrO9EUmVa6SIrJva?dl=0 does not have the code. The folder is empty. ---after author feedback and reading other reviews--- The authors have addressed some of my concerns: projection via imitation learning is easier, and provided the code and a video comparing with NDPS. The authors did not address my question on evaluation well. Although TORCS tasks are important, a single environment is not enough for evaluation. In particular, when can IPPG learn effectively via imitation learning can not be established via TORCS itself. Online imitation learning may be hard in many application domains. Can you leverage offline demonstrations? As a result, the paper still needs lots of improvement. However, I am willing to increase my rating to 6 given some of my concerns are addressed.

Reviewer 2



Very nice work. Some minor comments: 1. As in so many NIPS and ICML papers I review these days, the scholarship is somewhat shoddy. There's a vast literature of work on mirror descent based RL, including mirror-descent based safe policy gradient (Thomas et al, NIPS 2013), mirror-descent based sparse Q-learning (Mahadevan and Liu, UAI 2012), mirror-descent based gradient TD (Liu et al, UAI 2015, JAIR 2019), and many more articles in this vein. Please do a more thorough literature review. 2. From the results of Thomas et al., NIPS 2013, one can reinterpret your learning algorithm (Section 3) as doing natural gradient updates. Please read Thomas et al., and summarize your ideas from this perspective. 3. Following Mahadevan and Liu's work on sparse Q-learning, can you provide any sparsity guarantees on your learned policies (which would help in their interpretability). 4. Table 1 needs to be augmented with some of these earlier mirror-descent based approaches.

Reviewer 3



This paper addresses the problem of learning programmatic policies, which are structured in programmatic classes such as programming languages or regression trees. To this end, the paper proposes a "lift-and-project" framework (IPPG) that alternatively (1) optimizes a policy parameterized by a neural network in an unconstrained policy space and (2) projects the learned knowledge to space where the desired policy is constrained with a programmatic representation. Specifically, (1) is achieved by using deep policy gradient methods (e.g. DDPG, TRPO, etc.) and (2) is obtained by synthesizing programs to describe behaviors (program synthesis via imitation learning). The experiments on TORCS (a simulated car racing environment) show that the learned programmatic policies outperform the methods that imitate or distill a pre-trained neural policy and DDPG. Also, they exhibit stronger generalization ability and better verifiability (e.g. verifying the smoothness). I believe this work explores a promising direction for learning programmatic policies yet I am not entirely convinced by the experiments. I am will to raise the score if the rebuttal addresses my concerns/questions. [Strengths] *motivation* - The motivation for alternating between learning a neural policy and a programmatic policy to avoid obtaining a suboptimal programmatic policy from distilling a pre-trained neural policy is convincing and intuitive. *novelty & technical contribution* - I believe the idea of iteratively learning a programmatic policy with its neural counterpart is novel. This paper presents an effective way to implement this idea. - The provided theoretical analyses on the convergence and the overall performance are helpful. *clarity* - The paper gives clear descriptions in both theoretical and intuitive ways. The notations, formulations, and theorem are well-explained. *experimental* - The presentations of results are clear. - The proposed framework (IPPG) ----- outperforms the methods (VIPER and NDPS) that distill fixed neural policies and achieve comparable results compared to a neural policy learned using DDPG. ----- progressively improves the performance of a given sub-optimal programmatic policy (PRIOR). ----- exhibits stronger generalization ability (compared to DDPG): learn from a track and generalize to another. [Weaknesses] *assumption* - I am not sure if it is safe to assume any programmatic policy can be parameterized by a vector \theta and is differentiable in \theta. (for Theorem 4.2) *initial policy* - In all the experiments (TORCS, MountainCar, and Pendulum), the IPPG polices improve upon the PRIOR. It is not clear if IPPG can learn from scratch. Showing the performance of IPPG learning from scratch would be important to verify this. - Can IPPG be initialized with a neural policy? It seems that it is possible based on Algorithm 1. If so, it would be interesting to see how well IPPG work using a neural policy learned with DDPG instead of PRIOR. Can IIPG improve upon DDPG? *experiment setup* - It is mentioned that "both NDPS and VIPER rely on imitating a fixed neural policy oracle" (L244). What is this policy oracle? Is this the policy learned using DDPG shown in the tables? If not, what's the performance of using NDPS and VIPER to distill the DDPG policies? - It would be interesting to see if the proposed framework works with different policy gradient approaches. *experiment results* - How many random seeds are used for learning the policies (DDPO and IPPG)? - What are the standard deviation or confidence intervals for all performance values? Are all the tracks deterministic? Are the DDPG policies deterministic during testing? - It would be better if the authors provided some videos showing different policies controlling cars on different tracks so that we can have a better idea of how different methods work. *reproducibility* - Some implementation details are lacking from the main paper, which makes reproducing the results difficult. It is not clear to me what policy gradient approach is used. - The provided dropbox link leads to an empty folder (I checked it on July 5th). *related work* - I believe it would be better if some prior works [1-5] exploring learning-based program synthesis frameworks were mentioned in the paper. *reference* [1] "Neuro-symbolic program synthesis" in ICLR 2017 [2] "Robustfill: Neural program learning under noisy I/O" in ICML 2017 [3] "Leveraging Grammar and Reinforcement Learning for Neural Program Synthesis" in ICLR 2018 [4] "Neural program synthesis from diverse demonstration videos" in ICML 2018 [5] "Execution-Guided Neural Program Synthesis" in ICLR 2019 ----- final review ----- After reading the other reviews and the author response, I have mixed feelings about this paper. On one hand, I do recognize the importance of this problem and appreciate the proposed framework (IPPG). On the other hand, many of my concerns (e.g. the choices of initial policy, experiment setup, and experiment results) are not addressed, which makes me worried about the empirical performance of the proposed framework. To be more specific, I believe the following questions are important for understanding the performance of IPPG, which remain unanswered: (1) Can IPPG learn from scratch (i.e. where no neural policy could solve the task that we are interested in)? The authors stated that "IPPG can be initialized with a neural policy, learned for example via DDPG, and thus can be made to learn" in the rebuttal, which does not answer my question, but it is probably because my original question was confusing. (2) Can IPPG be initialized with a neural policy? If so, can IPPG be initialized with a policy learned using DDPG and improve it? As DDPG achieves great performance on different tracks, I am just interested in if IPPG can even improve it. (3) How many random seeds are used for learning the policies (DDPO and IPPG)? What are the standard deviation or confidence intervals for all performance values? I believe this is important for understanding the performance of RL algorithms. (4) What is the oracle policy that NDPS and VIPER learn from? If they do not learn from the DDPG policy, what is the performance if they distill the DDPG policy. (5) Can IPPG learn from a TPRO/PPO policy? While the authors mentioned that TRPO and PPO can't solve TORCS tasks, I believe this can be verified using the CartPole or other simpler environment. In sum, I decided to keep my score as 5. I am ok if this paper gets accepted (which is likely to happen given positive reviews from other reviewers) but I do hope this paper gets improved from the above points. Also, it would be good to discuss learning-based program synthesis frameworks as they are highly-related.

[Author Response · NeurIPS 2019]

We appreciate the reviewers' thoughtful comments. Due to the space limit, we only address the most salient con-
cerns/questions in the reviews. We will do our best to address the reviewers' other points in the paper's next version.

**Global responses to some common concerns among reviewers**

*Relationship to prior work (R2, R3)*: While mirror descent has been explored in RL settings before, our paper stands out
from all existing work in making three key innovations: (i) using a hybrid representation, a combination of a program
selected from a generic programming language and a neural network, for policies; (ii) implementing the projection
operator using combinatorial program synthesis; and (iii) a thorough analysis of the consequences of doing gradient
descent only in the neural component of hybrid policies, which introduces bias in the gradient operator.

To maintain focus on our main contributions, we used relatively basic versions of the two key components of the IPPG
meta-algorithm: mirror descent and program synthesis. However, each of these components can be improved further in
future work. In particular, like some of the papers pointed out by R2, one can use extensions of mirror descent such
as composite objective mirror descent [Duchi et al., JMLR11]. One can also implement the projection operator using
learning-accelerated program synthesis techniques, such as those pointed out by R3. We will include further discussion
of these possible directions in the final version.

*Incorrect link to code and reproducibility (R1, R3)*: We most sincerely apologize for the error! We accidentally created
two Dropbox folders during submission and uploaded the code to the incorrect folder (due to over-excitement with
our paper :-)). We have verified that the code is now available at the included link. We have also provided a Docker
container with all dependencies and the TORCS simulator installed, to ease reproducibility.

**Responses to other specific questions/concerns**

*R1 on why projection is easier than direct search in program space*: Intuitively, this is because we frame projection as an
imitation learning (IL) problem to imitate a neural net, in contrast to the full RL problem of searching over programmatic
space. It is widely acknowledged that IL tends to be easier than general RL, which needs to solve planning under
possibly long horizon (cf. Deeply AggreVaTeD paper by [Sun et al., ICML18]). This intuition is corroborated by
related work on programmatic RL [Verma et al., ICML18], which shows that direct search over programs often fails to
meet basic performance objectives (for example having a TORCS car finish a lap in any amount of time).

*R1 on our benchmarks and baselines being simple*: We respectfully disagree regarding the difficulty level of the
TORCS task, our primary benchmark. TORCS has several race tracks, many of which are challenging domains for RL
due to the continuous action space and long horizon. TORCS is also among the few standard benchmarks that allow
studying generalization properties, i.e., where the training and testing environment can differ. Of the popular policy
gradient algorithms, only DDPG has been shown to complete laps in TORCS in the tracks that we considered. Two
other algorithms, TRPO and PPO, cannot find policies capable of completing laps in most of these tracks [Cheng et al.,
ICML19]. In Table 1 in the paper, we show that IPPG learns to drive on some tracks (Ruudskogen and Alpine-2) where
even DDPG fails to learn a sensible policy. Thus, we do compare with (and show an improvement over) state-of-the-art
deep RL. Finally, we compare IPPG with VIPER and NDPS, the most relevant techniques for programmatic RL.

*R3 on assumption for Theorem 4.2*: This simplifying assumption is contained to Section 4.2 of the paper, for the purpose
of providing a clean finite-sample analysis under vanilla policy gradient. The subsequent Section 4.3 addresses the
setting when this assumption does not hold. Our analysis and experiments do not require this assumption in general.

*R3 on initial policy*: IPPG can be initialized with a neural policy, learned for example via DDPG, and thus can be made
to learn "from scratch". This amounts to running the program synthesis projection as a pre-training step (Lines 4-6 in
Alg. 1). On most tracks, this gives results comparable to those we have presented. However, note that IPPG with a (very
simple) hand-crafted prior can sometimes finish tracks on which DDPG fails. Naturally, this is not possible when we
invoke the program synthesis projection during pre-training. Also, using a prior results in "safer" training (Figure 5).

*R3 on experimental setup*: In principle, IPPG and its theoretical analysis do not depend on the particular policy gradient
approach. However, there is a huge empirical difference between different policy gradient algorithms on TORCS. PPO
and TRPO actually cannot find policies capable of completing laps in most TORCS tracks we considered (also noted in
some previous papers). For this reason, we did not try using them as a component of IPPG in this paper.

*R3 on experimental results & reproducibility*: We have added a video for the TORCS benchmark, comparing IPPG and
NDPS at 4 snapshots during training, to the Dropbox folder for the code. The video shows that IPPG is safer during
training, shows better iterative improvements, and recovers more gracefully from crashes than the NDPS agent. The
tracks are deterministic, but the initial state (starting position) of the car in each race is chosen randomly from a fixed
set. While we did not calculate confidence bounds in the submitted version, a key benefit of hybrid policies (over deep
RL) is that they lead to lower-variance learning [Cheng et al., ICML19], and as a result, IPPG is likely to have tighter
confidence bounds than DDPG. We will substantiate this point using concrete numbers in the final version of the paper.

[Meta-Review · NeurIPS 2019]

While the reviewers generally support acceptance, some concerns remain. We strongly encourage the authors to consider and address the concerns raised by the reviewers, as there remains room for improvement. While the paper is borderline due to these concerns, it falls on the side of acceptance due to the general support and strong support from reviewer 2.